# CRISPR-dCas13-tracing reveals transcriptional memory and limited mRNA export in developing zebrafish embryos

Youkui Huang[1], Bao-Qing Gao[2], Quan Meng[3,4], Liang-Zhong Yang[1], Xu-Kai Ma[5], Hao Wu[1], Yu-Hang Pan[1], Li Yang[5], Dong Li[3,4] and Ling-Ling Chen[1,6,7*]

*Correspondence:
linglingchen@sibcb.ac.cn

[1] State Key Laboratory of Molecular Biology, Shanghai Key Laboratory of Molecular Andrology, CAS Center for Excellence in Molecular Cell Science, Shanghai Institute of Biochemistry and Cell Biology, University of Chinese Academy of Sciences, Chinese Academy of Sciences, 320 Yueyang Road, Shanghai, China
Full list of author information is available at the end of the article

## Abstract

**Background:** Understanding gene transcription and mRNA-protein (mRNP) dynamics in single cells in a multicellular organism has been challenging. The catalytically dead CRISPR-Cas13 (dCas13) system has been used to visualize RNAs in live cells without genetic manipulation. We optimize this system to track developmentally expressed mRNAs in zebrafish embryos and to understand features of endogenous transcription kinetics and mRNP export.

**Results:** We report that zygotic microinjection of purified CRISPR-dCas13-fluorescent proteins and modified guide RNAs allows single- and dual-color tracking of developmentally expressed mRNAs in zebrafish embryos from zygotic genome activation (ZGA) until early segmentation period without genetic manipulation. Using this approach, we uncover non-synchronized de novo transcription between inter-alleles, synchronized post-mitotic re-activation in pairs of alleles, and transcriptional memory as an extrinsic noise that potentially contributes to synchronized post-mitotic re-activation. We also reveal rapid dCas13-engaged mRNP movement in the nucleus with a corralled and diffusive motion, but a wide varying range of rate-limiting mRNP export, which can be shortened by Alyref and Nxf1 overexpression.

**Conclusions:** This optimized dCas13-based toolkit enables robust spatial-temporal tracking of endogenous mRNAs and uncovers features of transcription and mRNP motion, providing a powerful toolkit for endogenous RNA visualization in a multicellular developmental organism.

**Keywords:** Zygotic microinjection, Modified gRNAs, De novo transcription, Post-mitotic, mRNP export, Developmental embryos

## Background

Gene expression is essential in all organisms, involving seamlessly coordinated steps of RNA transcription, splicing, export, translation, and degradation. RNA imaging techniques have been used to dissect these molecular events in a dynamic

view [1, 2]. Single-molecule fluorescence in situ hybridization (smFISH) has been well-established to visualize RNAs in fixed cells [3, 4], but methods for understanding transcription dynamics and mRNA export at a high spatial and temporal resolution in living cells, in particular, in vivo are still limited.

The MS2-MCP system uses the coat protein (MCP) of bacteriophage MS2 that binds tandem RNA stem-loops tagged to RNAs of interest. This system has been widely used to analyze RNA dynamics in bacteria, mammalian cells, and mice [5–8]. Similarly, the PP7-PCP system provides another visualizing tool for RNA tracking [9], and in combination with MS2-MCP enables dual-color RNA visualization [10]. Molecular beacons are a type of RNA aptamers that bind chemical fluorophores to visualize RNAs [11, 12]. In addition, the prokaryotic adaptive immune system CRISPR-Cas based on RNA-targeting Cas9 (RCas9) uses a mismatched protospacer adjacent motif as part of a complementary oligonucleotide (a modified PAMmer DNA) to target RNA and can image endogenous RNAs [13, 14]. More recently, the catalytically dead (d) RNA-guide and RNA-targeting RNases (known as the Cas13 family) fused to fluorescent proteins have been used to track mRNAs and/or locally enriched long noncoding RNAs in human cells [15–17]. These CRISPR-Cas-based technologies, especially the Cas13 system owing to its natural capability to recognize single-stranded RNAs, provide a simple and less time-consuming toolkit to visualize RNAs without genetic manipulation, compared to the widely used MS2-MCP system. However, the sensitivity and robustness of the CRISPR-dCas13 system still need to be improved to track endogenous mRNAs, and its utility in developing organisms has remained unexplored.

Tethering multiple MS2 aptamers to a reporter (e.g., *lacZ*, *fluorescent protein*), or ectopic or endogenous genes has enabled the observation of transcriptional burst [18–21], transcriptional memory inheritance during mitosis [22–25], and correlation of inter-allelic transcription [26]. However, it is a challenge to apply the MS2-MCP system to observe transcriptional profiles at endogenous loci in multicellular organisms, with a few studies showing transcriptional bursts and long-range gene regulation at the MS2-engineered loci in *Drosophila* embryos [27–30]. Moreover, it remains to be addressed how stochastic noise affects transcriptional fluctuations of inter-alleles from one cell cycle to another during the development of a multicellular organism. Thus, developing a robust non-genetic CRISPR-dCas13 system will be of importance to study de novo transcription and post-mitotic transcriptional re-activation of endogenous alleles in a multicellular organism.

As soon as transcribed, pre-mRNAs recruit RNA-binding proteins and are packaged into ribonucleoprotein (mRNP) particles [31, 32]. mRNPs are licensed for export by recognizing the transcription-export (TREX) complex, consisting of ALYREF, DDX39B, and the THO subcomplex [33, 34], followed by loading of the mRNP transport receptor, NXF1-NXT1 [35, 36]. Using MS2-tagged ectopic mRNAs, it was found that export-competent mRNPs move through the interchromatin space by simple diffusion [37, 38], and finally interact with nuclear pore complexes (NPCs) to translocate via NPC's central channel to the cytoplasm on the timescale of milliseconds [38–41]. The nuclear trafficking and export of genetically untagged mRNPs, and factors controlling the motion have remained to be explored.

Zebrafish (*Danio rerio*) embryos undergo extra-uterine development with optical transparency. We aimed to develop a robust CRISPR-dCas13 RNA imaging system to address questions concerning the dynamics and regulation of endogenous RNA transcription and export in these embryos. Screening of a panel of CRISPR-dCas13s combined with modified gRNAs enabled direct visualizations of mRNAs after ZGA to 15 cell cycles in enveloping layer (EVL) cells with a single zygotic injection of purified dCas13-fluorescent protein and modified gRNA. We observed highly heterogeneous de novo gene expression between individual pairs of alleles, which becomes synchronized during post-mitotic transcriptional re-expression, revealing inherited transcriptional states as extrinsic noise during embryogenesis. Furthermore, dCas13-engaged mRNPs move rapidly within the nucleus in a corralled and diffusive manner. Intriguingly, the nucleocytoplasmic export of such mRNPs via different diffusion patterns varied widely in export times throughout NPCs, leading to arrested mRNPs in the nucleus. Enhanced expression of Alyref or Nxf1 rescued the export of nuclear retained mRNPs.

## Results

### Establishing CRISPR-dCas13-mediated RNA labeling in zebrafish embryos

To develop a user-friendly CRISPR-dCas13 system for RNA labeling in zebrafish embryos, we examined a panel of dCas13 proteins to label ectopically expressed RNAs (Additional file 1: Fig. S1a). We first constructed *β-actin-48× GCN4* plasmid expressing *48× GCN4* elements [17], which encodes the GCN4 peptide epitope of the SunTag system [42], as the target of dCas13/gRNA complexes (Fig. 1a). For screening, each dCas13 protein was fused with enhanced green fluorescent protein (EGFP), the SV40 nuclear localization signal (NLS) at its N-terminus, and the Nucleoplasmin NLS at the C-terminus (abbreviated as dCas13-EGFP) to ensure nuclear dCas13 localization for RNA labeling (Additional file 1: Fig. S1b). Each dCas13-EGFP was purified (Additional file 1: Fig. S1c-e) and assembled with in vitro transcribed gRNAs targeting *48× GCN4* (*gGCN4*) (Additional file 2: Table S1) at a molar ratio 1:1.5 (for example, 5.6 μM complex, dPspCas13 protein: gRNA = 900 ng/μL: 160 ng/μL) (Fig. 1a). Different dCas13 proteins could be efficiently assembled with gRNAs in vitro, as shown by dPspCas13b/gRNA [17] and dRfxCas13d/gRNA [16] moving more quickly than the dCas13 protein alone on native PAGE gels (Additional file 1: Fig. S1f,g).

Next, we co-injected 5.6 μM pre-assembled dCas13-EGFP/gRNA complexes with 5.9 nM *β-actin-48× GCN4* plasmid into 1-cell stage zebrafish embryos (Fig. 1a). It is known that ectopic expression driven by the *β-actin* promoter begins after ZGA at about 3.5 h post fertilization (hpf) [44]. The *48× GCN4* RNA is highly expressed at 6 hpf, which is a time window to screen potentially active CRISPR-dCas13 systems for RNA labeling in fish embryos. We examined EGFP signals in the nucleus and evaluated whether these signals were indeed dCas13-labeled RNAs by single-molecule (sm)FISH probes to detect the ectopically expressed *48× GCN4* RNAs (Fig. 1a). Among all eight examined dCas13 proteins [45] (see also "Methods"), only the injection of dPspCas13b and its gRNAs resulted in specific EGFP puncta signals (white arrowheads shown, Additional file 1: Fig. S2a-h), as validated by smFISH in fixed cells that were colocalized with the dPspCas13b-labeled green signals (Additional file 1: Fig. S2i). The other seven dCas13 proteins did not show specific signals (Additional file 1: Fig. S2b-h). For example, injection

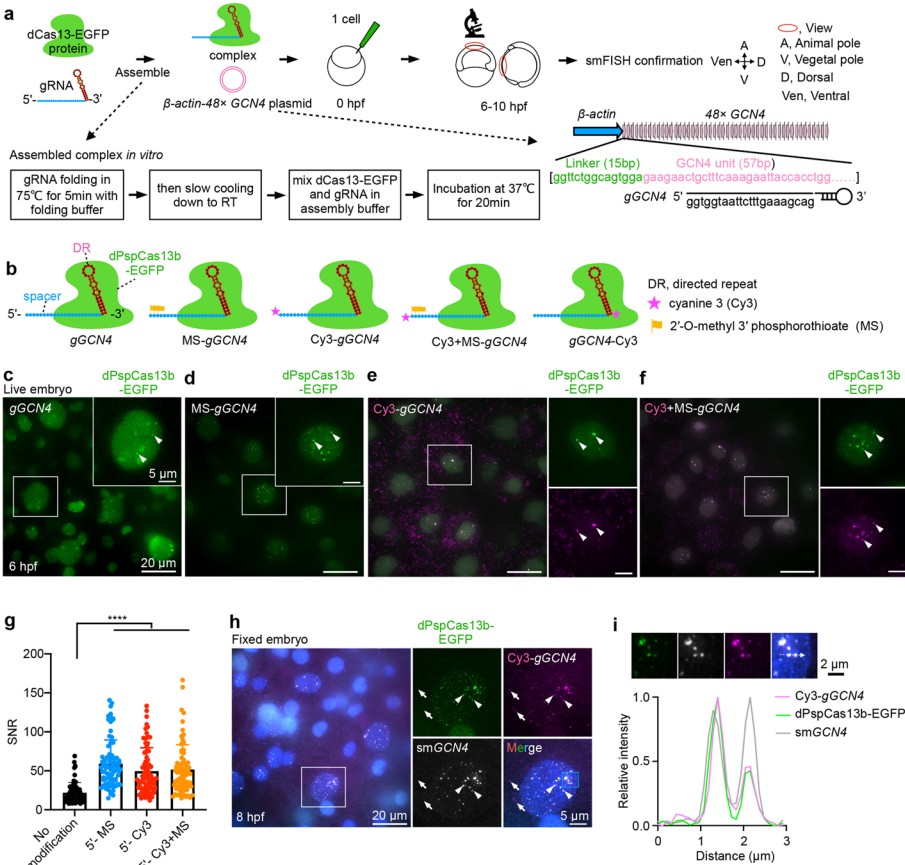

**Fig. 1** Imaging ectopic *GCN4* RNA repeats by CRISPR-dCas13 in zebrafish embryos. **a** A schematic screen to identify robust CRISPR-dCas13 systems to visualize RNAs in zebrafish embryos. 5.6 µM dCas13-EGFP protein assembled with 8.4 µM in vitro transcribed gRNA (*gGCN4*) at a molar ratio 1:1.5, plus 5.9 nM (50 ng/µL) *β-actin-48× GCN4* plasmid were injected into zebrafish embryos at the 1-cell stage. For some lower stability dCas13 constructs, we increased the concentration (see "Methods"). After injection, fluorescent signals in the nucleus were detected at 6–10 hpf (hours post fertilization). The CRISPR-dCas13 detected signals were confirmed by smFISH. *β-actin-48× GCN4*, zebrafish *β-actin* promoter derives *48× GCN4* expression. **b** A schematic view of chemically modified gRNA. Cyanine 3 (Cy3) and 2'-O-methyl 3' phosphorothioate (MS) modifications at the 5' and/or 3' end of *gGCN4* were used, including MS-*gGCN4*, Cy3-*gGCN4*, Cy3+MS-*gGCN4* and *gGCN4*-Cy3. The gRNA of dCas13b consists of a 5' spacer sequence (blue region) and a 3' direct repeat sequence (DR, red region) [43]. **c–f** Chemically modified gRNA improves CRISPR-dCas13 in RNA labeling. Representative images of CRISPR-dPspCas13b system-labeled *48× GCN4* with *gGCN4* (no modification, **c**), MS-*gGCN4* (**d**), Cy3-*gGCN4* (**e**), and Cy3+MS-*gGCN4* (**f**) at 6 hpf in live embryos, respectively. White arrowheads indicate labeled signals (**c–f**) and colocalization signals between Cy3 and EGFP (**e, f**). **g** SNR statistics of the *48× GCN4* signals labeled by unmodified and chemically modified of *gGCN4* with dPspCas13b-EGFP. Data from about 10 embryos in each group, $n =$ 71, 80, 73, 74 cells. **h** smFISH confirms the CRISPR-dPspCas13b system targeting *48× GCN4* at 8 hpf in fixed embryos. White arrowheads indicate colocalization of *GCN4* smFISH, Cy3-*gGCN4*, and dPspCas13b-EGFP. White arrows indicate non-specific signals. Blue box, zoomed in **i**. **i** Analysis of the blue box region in **h**. Line scan of the relative fluorescence intensity of signals (white dot line arrow in upper panel) shows the colocalization of Cy3-*gGCN4*, dPspCas13b-EGFP, and sm*GCN4* FISH. Scale bar 2 µm. In **c-f** and **h**, data scale bar 20 µm; the white box indicates magnified area, scale bar 5 µm. In **g**, data is represented as mean ± SD, unpaired two-tail Student's *t* test, **** $p<$ 0.0001

of dPguCas13b and dMisCas13b complexes resulted in formation of non-specific EGFP aggregates (magenta arrows, Additional file 1: Fig. S2f,g), which were also observed upon the injection of dPspCas13b (magenta arrows, Additional file 1: Fig. S2a). Such non-specific aggregates were larger than real signal puncta (white arrowheads, Additional file 1:

Fig. S2a,j) and localized to nucleoli (Additional file 1: Fig. S2k), thus could be easily separated from real RNA signals. Injection of dRfxCas13d, dBba2Cas13b, and dPba3Cas13b did not produce detectable EGFP puncta signals likely due to the using of inefficient unmodified gRNAs (see below), while dHgm4Cas13b and dHgm6Cas13b complexes appeared unstable in zebrafish embryos (Additional file 1: Fig. S2l), thereby showing no signals (Additional file 1: Fig. S2b-e, h, l).

For stable dPspCas13b-EGFP/gRNA complexes (Additional file 1: Fig. S2l), we observed that EGFP signals were localized in the nucleus and passed to each cell after division during embryo development. Such signals maintained sufficient brightness for imaging across 15 cell cycles with over 30,000 cells in one embryo (Additional file 1: Fig. S2m).

### Chemically modified gRNAs enhance the robustness of CRISPR-dCas13 RNA labeling

gRNAs modified by 2′-O-methyl 3′phosphorothioate (MS) improved CRISPR-Cas9-mediated genome editing efficiency [46] and CRISPR-RfxCas13d-mediated RNA knockdown [47] in human cells. Fluorescent gRNA and dRfxCas13d can target RNA transcription at an engineered LacO-repeated DNA locus in cells [16]. We asked whether chemical modifications of gRNAs could improve dCas13-mediated RNA labeling in developing embryos.

A gRNA is composed of the sequence targeting an RNA of interest (spacer) and the direct repeat (DR) being recognized by Cas13 to form an effector complex [43]. We introduced MS and cyanine 3 (Cy3), respectively, to the 5′ end spacer and the 3′ end DR of *gGCN4* (Fig. 1b; Additional file 2: Table S1). Modified *gGCN4s* were individually assembled with purified dPspCas13b-EGFP, and then co-microinjected with the *β-actin-48× GCN4* plasmid into 1-cell stage fish embryos. Cy3 and/or MS modifications at the 5′ end spacer of *gGCN4* showed dramatically increased EGFP puncta signals and enhanced SNR, compared to Cy3 at the 3′ end DR or the control unmodified gRNA (Fig. 1c–g and Additional file 1: Fig. S3a, b). Importantly, smFISH confirmed that these EGFP puncta signals were *48× GCN4* RNAs (Fig. 1h, i and Additional file 1: Fig. S3c,d). However, cytoplasmic Cy3 puncta were non-specific, which did not colocalize with *GCN4* using smFISH (Fig. 2h). The complexes containing the 3′ end DR modified *gGCN4* completely failed to label ectopically expressed *48× GCN4* RNAs (Additional file 1: Fig. S3a). This could be due to altered structures of gRNAs by 3′ DR modification, thus suppressing dPspCas13b recognition. Importantly, although dRfxCas13d and dPguCas13b failed to label *48×GCN4* RNAs with an unmodified gRNA (Additional file 1: Fig. S2f, h), modified gRNAs enabled RNA detection by dRfxCas13d and dPguCas13b both in live and fixed embryos (Additional file 1: Fig. S3e-h; Additional file 3: Table S2). Thus, the spacer modification of gRNA can improve the RNA labeling capacity of CRISPR-dCas13 in zebrafish embryos.

### CRISPR-dCas13 enables visualization of endogenous RNA in zebrafish embryos

Next, we asked whether this optimized CRISPR-dCas13 system (Fig. 1) could label endogenous RNAs in zebrafish embryos. The targeted transcripts were chosen to meet two criteria: (1) they contain ~20 nt repeated motifs to allow multiple dCas13-gRNA complex targeting; (2) they are expressed in specific cell types during early zebrafish

(See figure on next page.)

**Fig. 2** Single- and dual-color labeling endogenous mRNAs using CRISPR-dCas13 systems. **a** A pipeline to identify endogenous transcripts for dCas13 targeting. See also Additional file 1: Fig. S4a, b. **b** A schematic of the *eppk1* locus, which contains two exons. The transcript of *eppk1* is predicted to be about 16,900 nt and exon 2 contains 12 repeated units of about 1000 nt each. Three modified *geppk1s* were designed to target each unit. **c** Representative images of *eppk1* expression examined by WISH at 5.3 hpf, shown as the lateral (left) and dorsal (middle) views. A schematic *eppk1* expression in enveloping layer (EVL) cells is shown on right. **d** Representative images of the CRISPR-dPspCas13b system-labeled *eppk1* at 6 hpf in live embryos. White arrowheads indicate the colocalization between dPspCas13b-EGFP and Cy3-*geppk1s*. **e** smFISH confirms the CRISPR-dPspCas13b system-labeled *eppk1* at 8 hpf in fixed embryos. White arrowheads indicate the colocalization of *eppk1* smFISH, dPspCas13b-EGFP and Cy3-*geppk1s*. In some cells, four transcription sites (two pairs of signal spots) could be detected at two alleles. Blue box, zoomed in **f**. **f** Analysis of the blue box region in **e**. Line scan of the relative fluorescence intensity of signals (white dot line arrow in upper panel) shows the colocalization of Cy3-*geppk1s*, dPspCas13b-EGFP, and *eppk1* smFISH. Scale bar 2 μm. **g–i** Representative images of the CRISPR-dRfxCas13d system-labeled *eppk1* (white arrowheads) with *geppk1s*-MS at 6 hpf in live embryos (**g**), confirmed by smFISH (white arrowheads) in **h** and **i**. See **d–f** for details. **j** Mean SNR statistics of *eppk1* signals labeled by Cy3-*geppk1s* with dPspCas13b-EGFP and *geppk1s*-MS with dRfxCas13d-EGFP. Data from four independent experiments. Of note, the spacer on the gRNA of PspCas13b and RfxCas13d is at the 5′ and 3′ end, respectively [43]. **k** A schematic view of CRISPR-dPspCas13b-EGFP and CRISPR-dRfxCas13d-mScarlet systems to label two endogenous mRNAs in dual color with MS-modified gRNAs. **l** Representative images of *muc5.1* mRNA (green arrowheads) and *eppk1* mRNA (white arrowheads) labeled by dPspCas13b and dRfxCas13d systems at 10 hpf in live embryos, respectively. **m** Representative images of *100537515* mRNA (green arrowheads) and *eppk1* mRNA (white arrowheads) labeled by dPspCas13b and dRfxCas13d at 10 hpf in live embryos, respectively. In **d, e, g,** and **h,** data scale bar 20 μm; the white box indicates magnified area, scale bar 5 μm. In (l and m), data scale bar 5 μm. In **j**, data is represented as mean ± SD, unpaired two-tail Student's *t* test, ns, not significant

development to evaluate the reliability of detected signals. To this end, we analyzed expressed transcripts from single-cell RNA-seq dataset during zebrafish embryogenesis [48]. From a total of 1383 transcripts, 15 were expressed from 4 to 24 hpf and contained at least eight repeats (Fig. 2a and Additional file 1: Fig. S4a, b; Additional files 4,5,6: Tables S3,4,5). Among these 15 candidates, *eppk1* contains 12 repeated units of about 1000 nt each in exon 2 (Fig. 2b). Single-cell RNA-seq (Additional file 1: Fig. S4a, b), whole mount in situ hybridization (WISH), and smFISH confirmed that *eppk1* was specifically expressed in the EVL (Fig. 2c and Additional file 1: Fig. S4c), which is a single epithelial sheet protecting the embryo [49, 50].

To label *eppk1*, we designed three gRNAs carrying optimized modifications on spacers, named Cy3-*geppk1* and MS-*geppk1* for dPspCas13b, and *geppk1*-MS for dRfxCas13d, in the exon 2 (Fig. 2b; Additional file 2: Table S1). Live embryo images showed that dPspCas13b and dRfxCas13d displayed EGFP puncta signals in the nucleus of nearly all EVL cells (Fig. 2d,g and Additional file 1: Fig. S5a), but not in other cell types (Additional file 1: S5b, c). Colocalization of dPspCas13b-EGFP, Cy3-*geppk1s*, and *eppk1* smFISH signals (Fig. 2d–f), as well as dRfxCas13d-EGFP and *eppk1* smFISH signals (Fig. 2g–i), confirmed that dPspCas13b-EGFP and dRfxCas13d-EGFP efficiently labeled *eppk1* mRNAs in EVL cells. However, dPguCas13b failed to label endogenous *eppk1* mRNAs (Additional file 3: Table S2). These nuclear spots of endogenous transcription sites of *eppk1* in each living and fixed EVL cell were fitted with Gaussian function (Fig. 2e, f, h, i and Additional file 1: Fig. S5d-f), indicating CRISPR-dCas13 systems efficiently target endogenous *eppk1* mRNAs at their sites of transcription.

Next, we optimized the assembly concentration of dCas13 proteins and gRNAs, aiming to achieve the best SNR (Additional file 1: Fig. S5g-p). Reduced ratios of gRNAs to

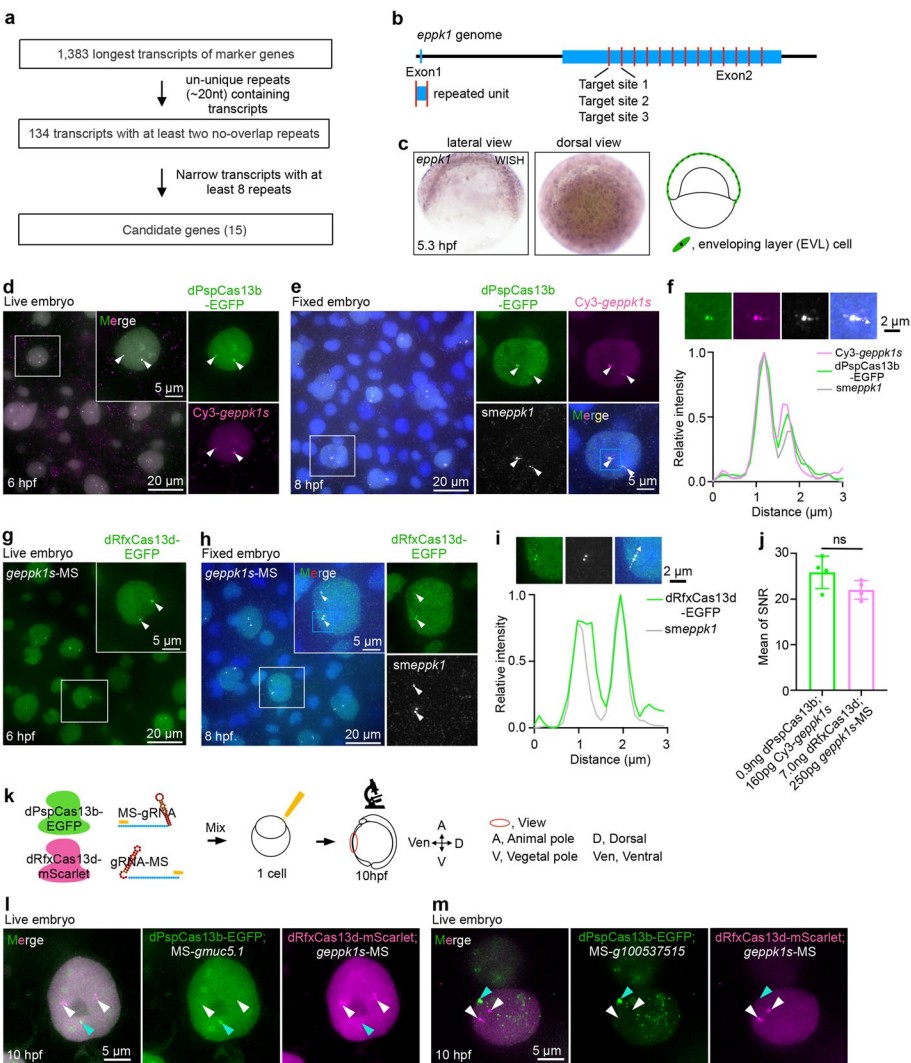

**Fig. 2** (See legend on previous page.)

dPspCas13b in the final dPspCas13b-EGFP/Cy3-gRNAs complexes (i.e., each embryo was injected with dPspCas13b-EGFP/Cy3-gRNAs, 0.9 ng/53 pg vs 0.9 ng/160 pg; Additional file 1: Fig. S5g-j) showed comparably robust RNA labeling efficiency (Additional file 1: Fig. S5k), whereas increased concentrations of dPspCas13b in the final RNP complexes resulted in enhanced background, and thereby reduced SNR (Additional file 1: Fig. S5g-k). In contrast, increased concentrations of dRfxCas13d improved the SNR of labeling signals (Additional file 1: Fig. S5l-p), even when using reduced concentrations of gRNAs (Additional file 1: Fig. S5m,o,p), suggesting that the dRfxCas13d protein is less stable than dPspCas13b and that a higher concentration of dRfxCas13d than dPspCas13b was warranted for RNA labeling in developing fish embryos.

Altogether, these optimization steps allowed us to apply complexes of dPspCas13b-EGFP/Cy3-gRNAs (0.9 ng/160 ng) and dRfxCas13d-EGFP/gRNAs-MS (7 ng/250 ng) for zygotic injection to achieve a comparable labeling ability (Fig. 2j). Of note, the CRISPR-dPspCas13b system showed high labeling efficiency and undetectable off-target rate at

examined transcription sites (Additional file 1: Fig. S5q, r). Notably, both dCas13 systems showed no obvious toxicity to embryo development with these optimized dosages, as all examined embryos were developmentally normal and *eppk1* expression remained unaltered (Additional file 1: Fig. S5s,t).

Interestingly, microinjection of the pre-assembled dCas13-EGFP/modified gRNA complex, or microinjection of the dCas13-EGFP/modified gRNA without a pre-assembly step (Additional file 1: Fig. S6a, see also "Methods") resulted in a similar *eppk1* labeling efficiency (Additional file 1: Fig. S6b-e). However, microinjection of mRNAs encoding the fused protein (Additional file 1: Fig. S6f) showed aggregated non-specific EGFP signals without detectable *eppk1* signals in EVL cells (Additional file 1: Fig. S6g, h). These attempts showed that dCas13 protein and gRNA need to be delivered simultaneously.

### Dual-color labeling of endogenous RNAs using orthogonal dCas13 systems

PspCas13b and RfxCas13d are two orthogonal CRISPR-dCas13 proteins. We thus attempted to visualize other transcripts expressed in EVL cells in addition to *eppk1* (Additional file 1: Fig. S4b). We synthesized 5′ spacer MS-modified gRNAs (MS-*gmuc5.1* and MS-*g100537515*) to target *muc5.1* and *100537515* (PubMed gene ID; Official full name *si:cabz01007794.1*) (Additional file 1: Fig. S7a, b; Additional file 2: Table S1). smFISH confirmed their expression at 9–10 hpf (Additional file 1: Fig. S7c), which is the time window we have chosen for living cell imaging by dPspCas13b-EGFP/MS-gRNA microinjection (Additional file 1: Fig. S7d). Live embryo images showed that dPsp-Cas13b-EGFP could label *muc5.1* (Additional file 1: Fig. S7e) and *100537515* (Additional file 1: Fig. S7h) transcripts with single gRNA individually screened from 4 to 7 gRNAs (Data not shown), validated by smFISH (Additional file 1: Fig. S7f, g, i, j). Next, we microinjected dPspCas13b-EGFP/MS-gRNA and dRfxCas13d-mScarlet/gRNA-MS to target either *muc5.1* and *eppk1*, or *100537515* and *eppk1*, respectively into 1-cell zygotes, and allowed their development to 10 hpf (Fig. 2k). We observed their transcription sites in dual colors in the nuclei of live embryos under these different combinations (Fig. 2l, m). Strikingly, the single gRNA-targeted *100537515* marked not only sites of transcription, but also transcribed RNAs in both nucleoplasm and cytoplasm (Additional file 1: Fig. S7i,g). Labeled transcription sites were confirmed by the colocalization between sm-*intron* FISH of pre-mRNA and dPspCas13b-EGFP-labeled signals (Additional file 1: Fig. S7k,l). Of note, dPspCas13b has also enabled ~ 47.3% of total nucleoplasmic *100537515* mRNAs with ~4.1% off-target rate (Additional file 1: Fig. S7m, n), while a much lower labeling of cytoplasmic *100537515* mRNAs (Data not shown).

### Non-synchronized de novo transcription in developing zebrafish embryos

Next, we examined dynamic transcription at endogenous loci in developing embryos by tracking de novo transcription of *eppk1* and *100537515* using the CRISPR-dPspCas13b system. To capture de novo transcription events during early embryo development, we applied time-lapse imaging beginning from 3.3 hpf at the early stage of ZGA with Olympus SpinSR confocal microscopy (Fig. 3a). One serial image stack was recorded every 2 or 5 min for hours to capture nascent mRNAs, and two transcription sites were analyzed by maximum intensity projection (Additional file 1: Fig. S8a-c).

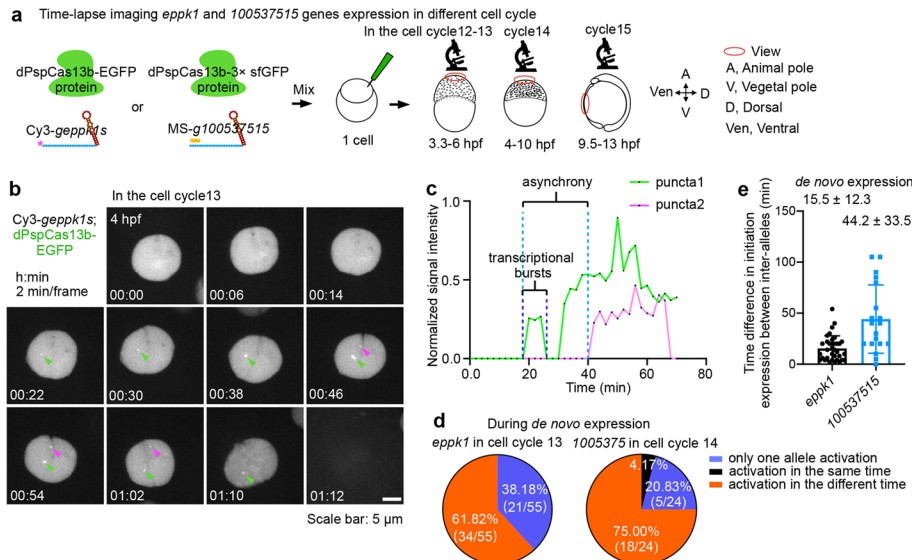

**Fig. 3** Non-synchronized de novo transcription in developing zebrafish embryos. **a** A schematic of time-lapse imaging of *eppk1* and *100537515* gene expressions in different cell cycles of developing embryos. About 70 image stacks were recorded every 2 min during cell cycle 12–13 from 3.3 to 6 hpf, or every 5 min during cell cycle 14 from 4 to 10 hpf, and cell cycle 15 from 9.5 to 13 hpf. The image stacks were 0.4 μm *z*-step distance and then were analyzed by maximum intensity projection. **b** Representative images of dPspCas13b-EGFP tracking de novo transcription of *eppk1* with Cy3-*geppk1s*, recorded every 2 min from 4 to 6 hpf during EVL of cell cycle 13, see also Additional file 7: Movie 3. Green arrowheads indicate puncta 1, referred to as the first detected initial transcription site in each cell throughout the study, and magenta arrowheads indicate puncta 2, the later initial transcription site in each cell throughout the study. **c** Non-synchronized de novo transcription of inter-alleles at *eppk1* in developing zebrafish embryos. Normalized intensity at the transcription sites recorded over time in **b**. Puncta 1 and 2 are indicative of the two transcription sites. Initial expression of puncta 1 and puncta 2 are asynchrony. Transcriptional burst shows switching a promoter from activated state to inactivated state. **d** Non-synchronized de novo transcription of EVL cells. Statistics of de novo transcription events of two genes *eppk1* (left) and *100537515* (right) in individual cell cycles (cell cycles 13 and 14, respectively), in which their de novo transcription is first detected. Three types of de novo transcription events are recorded throughout one cell cycle: (1) only one allele is transcribed (blue); (2) two alleles are transcribed at the same time (black); (3) two alleles are transcribed sequentially (orange). See also Additional file 1: Fig. S8f, g for original data. See also Fig. 3e for detail. **e** Time difference in initial transcription between individual inter-alleles of *eppk1* and *100537515* during de novo expression in cell cycles 13 and 14, respectively. See also Additional file 1: Fig. S8f, g for original data. Data are represented as mean ± SD

We first observed that *eppk1* initiated its transcription in EVL cells at cell cycle 13 at approximately 4 hpf (Additional file 1: Fig. S8f, h; Additional file 7: Movie 1) and sufficiently maintained its high SNR to 8 hpf (Additional file 7: Movie 5). *Eppk1* has two copies in the genome of diploid embryonic cells. We visualized the time course of both alleles labeled with dPspCas13b-EGFP/Cy3-*geppk1s* (Additional file 7: Movie 1). To quantify transcriptional behaviors, we manually tracked the two alleles and measured the fluorescence intensity in real time (Fig. 3b, c; Additional file 7: Movie 3; see also "Methods"). By analyzing transcriptional fluctuations of each allele of *eppk1* over time, we observed that one of the two alleles first initiated expression at around 20 min after 4 hpf (annotated as puncta 1), and the other allele was expressed later at approximately 42 min (annotated as puncta 2) in the same cell during de novo transcription of cell cycle 13 of EVL (Fig. 3c; Additional file 7: Movie 3), indicating that de novo transcriptional initiation of inter-alleles was asynchronous.

A similar de novo transcription pattern of *100537515* was also observed, starting at the cell cycle 14 of EVL cells after 6 hpf (Additional file 1: Fig. S8d, e, g, h; Additional file 7: Movies 2,4), indicating that the non-homologous de novo transcription of two sister alleles is not unique to *eppk1*. Our tracking results also showed varied levels of transcription followed by surges or bursts of expression at single alleles (Fig. 3c and Additional file 1: Fig. S8e). These fluctuations are known as transcriptional bursts involving random switching of promoters between active and inactive states reported in other model organisms [51]. We also detected one allele with two distinguishable signal spots (alleles are shown as green arrowheads) in close proximity in the later stage of the measured cell cycle (Additional file 1: Fig. S8d; Additional file 7: Movie 4) corresponding to transcription sites on sister chromatids [20].

To characterize inter- and intra-cell heterogeneous transcription events, we analyzed all traces of inter-alleles from *eppk1* and *100537515* in cell cycles 13 and 14 of EVL, respectively (Additional file 1: Fig. S8f,g). Among all observed EVL cells, a fraction of EVL cells (42.7%, 41 cells from total 96 cells) never initiated *eppk1* expression during cell cycle 13; but nearly all EVL cells expressed *100537515* during cell cycle 14. In activated EVL cells, 38.18 and 20.83% cells only activated one allele of *eppk1* and *100537515*, respectively, but expression of the other allele was never initiated (Fig. 3d). Quantification of the kick-off expression time of inter-alleles revealed a highly variable firing of initial transcription, especially for *100537515* (Fig. 3e). Altogether, these observations suggest non-synchronized de novo allelic transcription at EVL cells in developing embryos.

### Inherited transcriptional states of de novo transcription become synchronized post-mitosis

Mitotic changes imposed on the nuclear envelope, chromatin compaction, and intra-chromosomal and enhancer-promoter interactions lead to changes of many transcriptional events [52, 53]. We examined transcriptional profiles of *eppk1* and *100537515* re-activations in the next cell cycle. To this end, we manually tracked *eppk1* and *100537515* transcription sites of daughter cells (Fig. 4a, b and Additional file 1: Fig. S9a-d; Additional file 7: Movies 5,6), mothers of whom expressed *eppk1* in cell cycle 13 or *100537515* in cell cycle 14 of EVL cells. According to the nuclear morphology illustrated by EGFP fluorescence, we set zero minute as soon as the initiation of transcriptional re-activation upon re-establishing clear nuclei in daughter cells. Interestingly, on exit from mitosis, transcription of both *eppk1* and *100537515* was rapidly re-activated and quickly reached the peak of transcriptional activity for both alleles in individual single cells (Fig. 4a, b and Additional file 1: Fig. S9a, b; Additional file 7: Movies 7,8). In contrast to the de novo transcription cell cycle, a portion of EVL cells showed quick and synchronous re-activation of *eppk1* and *100537515* within ~10 min after mitosis (Additional file 1: Fig. S9c-e). Indeed, transcriptional sites 1 and 2 (puncta1 and 2) of both genes were re-activated at the same time in 20–30% EVL cells in the post-mitotic cell cycle, and single-allele re-activation events were nearly undetectable (Fig. 4c). Interestingly, quantification of these events showed a similar possibility of re-activation expression in alleles and between different genes within the examined post-mitotic cell cycles at *eppk1* and *100537515* loci (Fig. 4d and Additional file 1: Fig. S9e, f). Thus, the profiles of transcriptional initiation between de novo transcription and post-mitotic re-activation are

different, in which the asynchronous de novo transcription becomes more synchronized re-activation after cell division.

These observations prompted us to ask whether inherited transcriptional states can propagate from mother to daughter cells through cell division, thereby contributing to synchronous transcription in daughter cells, referred to as transcriptional memory [52, 53]. Indeed, we observed reduced timing in initial expression of inter-alleles and of daughter cells during post-mitotic re-activation, compared to inter-allelic de novo transcription in the same cells (Fig. 4e), which is a feature of transcriptional memory. As observed in *eppk1* locus, inter-allelic activation at the same time was only detected in descendant cells (19.44%) (Fig. 4c), but not in the previous de novo transcription cell cycle (Fig. 3d). Of note, the *100537515* gene had a significantly reduced timing in initial expression between inter-alleles during post-mitotic re-activation (post-mitotic re-activation, 11.9 ± 14.6 min vs de novo transcription, 44.2 ± 33.5 min) (Fig. 4e). Remarkably, the different timing in firing of re-activation between daughter cells of *eppk1* and

(See figure on next page.)

**Fig. 4** Inheritance of active transcriptional states propagating through cell division. **a** Representative images of dPspCas13b-EGFP tracking *eppk1* re-activation with Cy3-*geppk1s*, recorded every 5 min from 4.5 to 8 hpf in EVL of cell cycle 14. See also Additional file 7: Movie 7. Green arrowheads indicate puncta 1, and magenta arrowheads indicate puncta 2. **b** Synchronized transcription of inter-allelic *eppk1* in the cell cycle post the cell cycle having de novo transcription. Normalized signal intensity of transcriptional activity recorded over time in **a** in the EVL of cell cycle 14 in developing embryos. **c** Synchronized post-mitotic transcriptional re-activation of EVL cells. Statistics of the inherited active transcriptional sites of two genes *eppk1* (left) and *100537515* (right) during re-activation in the next cell cycles (the EVL of cell cycles 14 and 15, respectively). Data were traced from Additional file 1: Fig. S9c, d for original data. See also Fig. 4d for detail. **d** Time difference in initial transcription between individual inter-alleles of *eppk1*(left) and *100537515* (right) during transcription re-activation in the next cell cycles (the EVL of cell cycles 14 and 15, respectively). Data were traced from Additional file 1: Fig. S9c, d. See also Additional file 1: Fig. S9f. **e** Inherited active transcriptional states propagating through cell division. Statistics of the time difference in initial transcription between inter-alleles, and among pairs of sister cells for *eppk1* and *100537515* during de novo transcription and post-mitotic re-activation in different cell cycles. See also Figs. 3e, 4d and Additional file 1: Fig. S9f-h. **f, g** Rapid kinetics of post-mitotic transcriptional re-activation. Expression of both *eppk1* (**f**) and *100537515* (**g**) is more rapidly re-activated and reached a transcriptional plateau in the post de novo transcription cell cycle of EVL cells. Normalized intensity in all cells (*eppk1*, n= 55 in the cell cycle 13, 0 time point indicating 4 hpf; n= 36 in the cell cycle 14, 0 time point indicating 4.5 hpf. *100537515* n= 24 in the cell cycle 14, 0 time point indicating 6 hpf; n = 36 in the cell cycle 15, 0 time point indicating 9.5 hpf) at each time point across de novo expression (cell cycle 13 for *eppk1*, and cell cycle 14 for *100537515*) and post-mitotic transcriptional re-activation (cell cycle 14 for *eppk1*, and cell cycle 15 for *100537515*) cell cycles. Sum of puncta 1 and puncta 2 for one cell transcriptional activity at each time point. Of note, the non-synchronized transcription of *100537515* inter-alleles in cell cycle 14 can be used as a control for the synchronized transcription of *eppk1* inter-alleles in the same cell cycle, and vice versa. Data were traced from Additional file 1: Fig. S8f, g and S9c, d for original data. **h–k** Scatterplot of transcriptional output of pairs allelic *eppk1* in the same cells (**h, i**) and random pairs allelic *eppk1* from the different cells (**j, k**) during EVL cell cycles 13 (**h, j**) and 14 (**i, k**), respectively. Total RNA output was determined by summing the area under the time point traced at each allele (data from Additional file 1: Fig. S8f,9c). *r*: Pearson correlation coefficient, *ρ*: Spearman's rank correlation coefficient, *k*: Slope. The pink line in each panel indicates slope. **l, m** Increased inter-allelic correlations of *eppk1* (**l**) and *100537515* (**m**), respectively, during post-mitotic transcriptional re-activation, compared with those during de novo expression. Statistics of the Pearson correlation coefficient of all cells (*eppk1*, n = 55 in the cell cycle 13; n = 36 in the cell cycle 14. *100537515* n = 24 in the cell cycle 14; n = 36 in the cell cycle 15) at each time point. See also Additional file 1: Fig. S11e-h. Pearson correlation coefficient in real time. *eppk1*, *p* = 1.17e−06; *100537S515*, *p* = 0.01. Data were traced from Additional file 1: Fig. S8f, g and S9c, d for original data. **n** The transcriptional activity has no difference in random pairs of *eppk1* (left panel) and *100537515* (right panel) alleles from different cells between de novo expression and post-mitotic transcriptional re-activation, shown by Pearson correlation coefficient. *eppk1*, *p* = 0.85; *100537515*, *p* = 0.91. In **d–g**, data are represented as mean ± SD. In **d, e**, data unpaired two-tail Student's *t* test; ns, not significant; * *p*< 0.05, **** *p*< 0.0001. In **l–n**, data of the box, middle line is median; upper and lower horizontal lines are 25% and 75% quartiles respectively; the point is mean; unpaired two-tail Student's *t* test

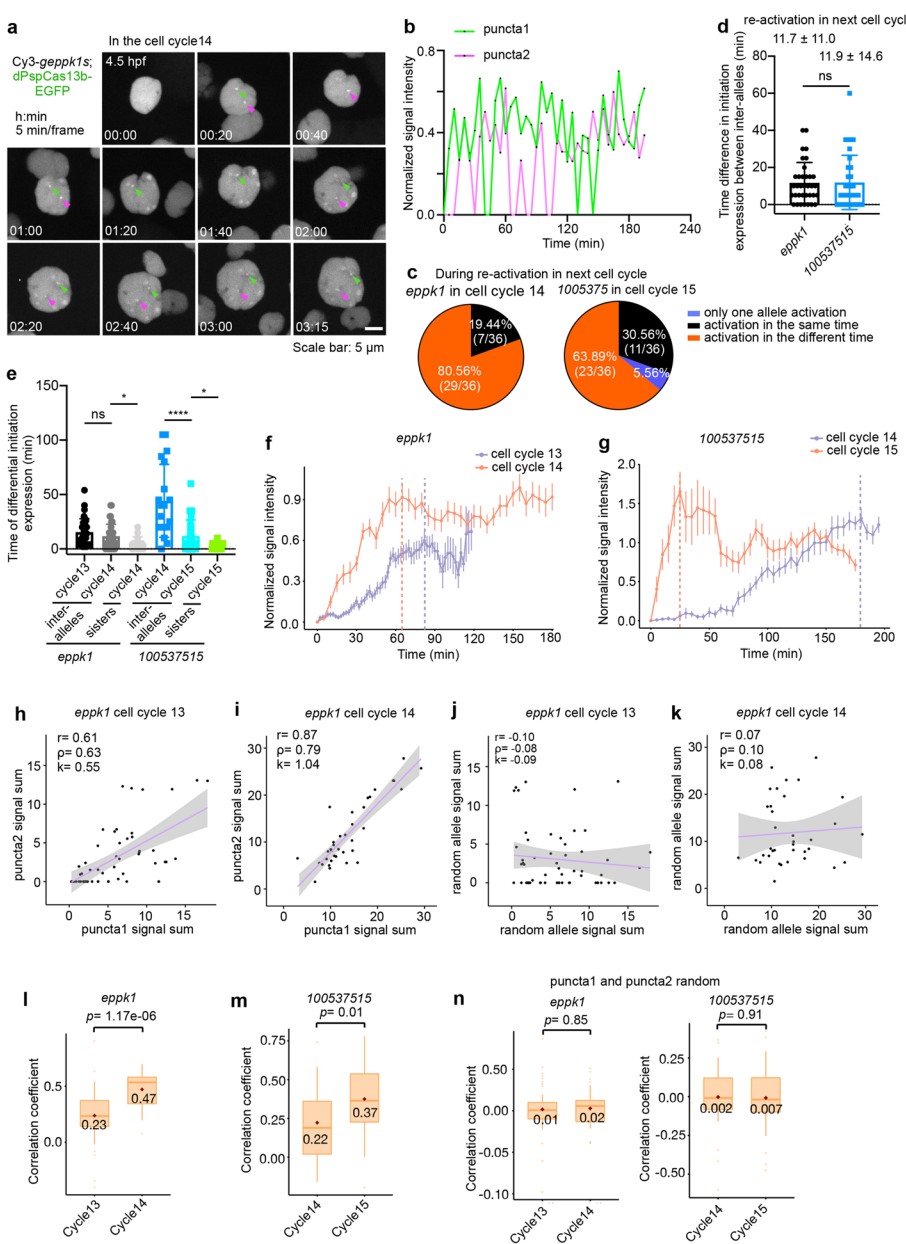

**Fig. 4** (See legend on previous page.)

*100537515* was consistently decreased (Fig. 4e and Additional file 1: Fig. S9g, h), which was similar to the memory daughters (8.5 ± 6.2 min) in *Drosophila* embryos [24].

Next, we quantified transcriptional profiles from the de novo transcription cell cycle to the post-mitotic cell cycle in EVL cells. De novo transcription activity of *eppk1* and *100537515* reached a plateau peak at 82 and 180 min, respectively (Fig. 4f,g), indicating a steady but slow increase in transcription of both alleles. In contrast to the slow kinetics associated with de novo transcription, transcriptional re-activation of both *eppk1* and *100537515* reached a plateau peak much more rapidly (65 and 25 min, respectively) in their respective post-mitotic cycles (Fig. 4f,g). In particular, at the *100537515* locus, dPspCas13b-3× sfGFP recruitment to the transcription site

(puncta 1, the first activation allele) during transcriptional re-activation was 7 times faster than that of de novo transcription (25 min during post-mitotic re-activation vs 180 min during de novo transcription). These observations are consistent with the finding that rapid post-mitotic transcriptional re-activation is mediated by transcriptional memory using the MCP-MS2 system [23]. Taken together, the observed kinetics of post-mitotic transcriptional re-activation indicated that transcription memory could allow the inheritance of transcriptional states from mother cells, which may be a potential mechanism to promote asynchronous de novo transcription to become synchronized after cell division in multicellular organisms.

### Transcriptional memory as an extrinsic noise may mediate post-mitotic inter-allele synchronous transcription

To ask how transcription memory may affect transcriptional fluctuations in real time, we plotted inter-allelic transcriptional activities over time to compare de novo transcription and post-mitotic re-activation. Differences in inter-allele transcriptional activities at both *eppk1* and *100537515* genes were observed. While the difference in transcription activity was slowly increased during de novo transcription (Additional file 1: Fig. S10a, c), it rapidly reached a peak during post-mitotic transcriptional re-activation (Additional file 1: Fig. S10b,d). These findings indicated that transcriptional fluctuations were reduced after mitosis, leading to more synchronous transcription between post-mitotic inter-alleles (Figs. 3 and 4a–g and Additional file 1: Fig. S8,9).

Both intrinsic and extrinsic noise contribute to transcriptional fluctuations [54]. While the intrinsic noise results from stochastic biochemical reactions in the process of gene expression that affects each allele of a gene independently, extrinsic noise results from fluctuations of cellular components, for example, cell cycle. To separate the contributions of these two sources of noise, we quantified the correlation of inter-alleles to compare changes of transcriptional fluctuations in two continuous cell cycles of gene expression by examining the intrinsic and extrinsic noises via plotting transcriptional outputs of one allele versus another in single cells (Fig. 4h–k). Cells distributed along the diagonal were attributed to extrinsic noise, referred to as "correlated," and those off the diagonal were attributed to intrinsic noise, referred to as "uncorrelated" transcriptional outputs of inter-alleles (Fig. 4h, i). Compared to the de novo transcriptional cell cycle 13 at *eppk1* (Fig. 4h), the transcriptional outputs of its two alleles in the same cells during post-mitotic re-activation (cell cycle 14, Fig. 4i) displayed a higher correlation. Concordantly, by plotting transcriptional activities of the puncta 1 (the transcribed allele detected first) against the puncta 2 (the other allele) in real time in the same cell (Additional file 1: Fig. S11e, f), the post-mitotic transcriptional activity of *eppk1* inter-alleles showed a more significantly positive correlation than the de novo transcription within the same cells (Fig. 4l). As controls, random pairs of alleles from different cells did not display a correlation (Fig. 4j, k, n). Similar transcriptional outputs at *100537515* were also observed from its de novo transcription cell cycle 14 to the post-mitotic re-activation cell cycle 15 (Fig. 4m, n and Additional file 1: Fig. S11a-d, g, h).

Together, this optimized CRISPR-dCas13-based RNA imaging system has captured transcriptional fluctuations of endogenous alleles in the same cells across two continuous cell cycles, showing that extrinsic noise likely dominated the uncorrelated intrinsic

variations during de novo transcription, contributing to the post-mitotic synchronized transcriptional outputs. Thus, transcriptional memory likely serves as an extrinsic noise to promote the switch from non-synchronized de novo transcription to synchronous transcription in developing embryos.

### Nuclear and cytoplasmic motions of dPspCas13b/100537515 mRNPs

The dynamics of mRNPs have been characterized using indirect measures (e.g., fluorescence reporter genes) [37] or at genetically manipulated loci in cell lines [39]. However, their motions without genetic tagging are limited in multicellular organisms. The optimized CRISPR-dCas13 RNA imaging system has enabled us to track single *100537515* mRNPs, providing a new perspective on the motion of endogenous mRNPs in developing embryos. It should be noted that this CRISPR-dCas13 system has enabled single-molecule efficiency, as revealed by the fact that per *100537515* mRNP contained ~11 copies of dPspCas13b-EGFP, similar to the predicted 12 target sites for dPspCas13b-EGFP binding (Additional file 1: Fig. S12a),

  We recorded the movement of the dPspCas13b-3× sfGFP engaged *100537515* mRNPs from 10 to 12 hpf by sequential imaging in live embryos at the rate of 100 images per second (Fig. 5a and Additional file 1: Fig. S12b), allowing the capture of the fast motion of mRNPs. Next, we characterized the movement of mRNPs by performing single-particle tracking on particles that remained in focus for a minimum of 20 consecutive frames ($\geq$ 0.2 s) (Additional file 1: Fig. S12c,d). The trajectories were analyzed by the mean square displacement (MSD) plotted over time $\triangle t$ (Additional file 1: Fig. S12e, right panel), which has been used to quantify kinetics of mRNP movement [37]. The observed

(See figure on next page.)

**Fig. 5** dPspCas13b/*100537515* mRNP export via different patterns, which are modulated by Alyref and Nxf1. **a** A schematic view of the CRISPR-dPspCas13b tracked *100537515* mRNP motion in the nucleus and cytoplasm by time-lapse imaging. **b** Representative images of dPspCas13b-engaged *100537515* mRNP export, recorded every 10 ms. Pom121-mScarlet marks nuclear pore complexes (NPCs). Right panel, maximum time projection (max. proj.) of the movie shows the trajectory of mRNP export. Green arrowheads indicate mRNPs, see also Additional file 7: Movie 14. **c** Time of dPspCas13b/*100537515* mRNP export displays a wide range. Left panel, graph depicting export time of all tracked mRNPs (*n*= 38) varies from 0.10 to 30 s. Right panel, graph shows the types of export events, including fast transport into cytoplasm less than 1 s (*n*=10) with directed transport via linearly traveling into cytoplasm (*n*=5), and slow transport into cytoplasm more than 1 s (*n*=23). **d** Graph depicting trajectory of slow transport event translocating from the nuclear envelope to the cytoplasm, with a significant dwell duration inside the nuclear envelope. **e** Two types of motions of mRNPs shown by the mean square displacement (MSD) of tracked export event in **d** versus time. The presence of two types of motions: diffusive movement (red line) in the cytoplasm and stationary diffusion (blue line) transporting through nuclear pore. **f** Time of dPspCas13b/*100537515* mRNP export displays a wider range. Export time requirements of tracked mRNPs adopt different interval time for imaging, including 0.05 s (*n* = 12 from 38 cells), 0.2 s (*n* = 9 from 37 cells), and 2 s (*n* = 13, from 49 cells), ranging from 0.3 to 180 s. **g** Alyref and Nxf1 promotes mRNP export in developing embryos. Representative images of smFISH of *100537515* mRNP distribution in the nucleus and cytoplasm after injected the CRISPR-dPspCas13b system with transport factors mRNA, *alyref* and *nxf1* at 10 hpf in fixed embryos. Scale bar 20 μm; white box indicates magnified area, scale bar 5 μm. **h** Statistical analyses show that Alyref and Nxf1 promotes mRNP export in developing embryos. Graph depicting nuclear/cytoplasm ratio of *100537515* mRNA in different group of examined EVL cells from about 10 developing embryos in **g**; *n* = 32, 33, 34 cells. **i** Alyref or Nxf1 overexpression shortens the time of dPspCas13b/*100537515* mRNP export. The mRNPs were recorded at 50 ms per frame resolution. Export events: Ctrl, *n* = 86 (from 231 cells); *alyref*, *n* = 142 (from 255 cells); *nxf1*, *n* = 112 (from 272 cells). Mann-Whitney test was used; center dotted line, median; upper and lower dotted lines, 25% and 75% quartiles; **** $p < 0.0001$. In **c, f**, data are represented as mean $\pm$ SEM. In **h**, data are represented as mean $\pm$ SD; unpaired two-tail Student's *t* test; **** $p < 0.0001$. Ctrl, control

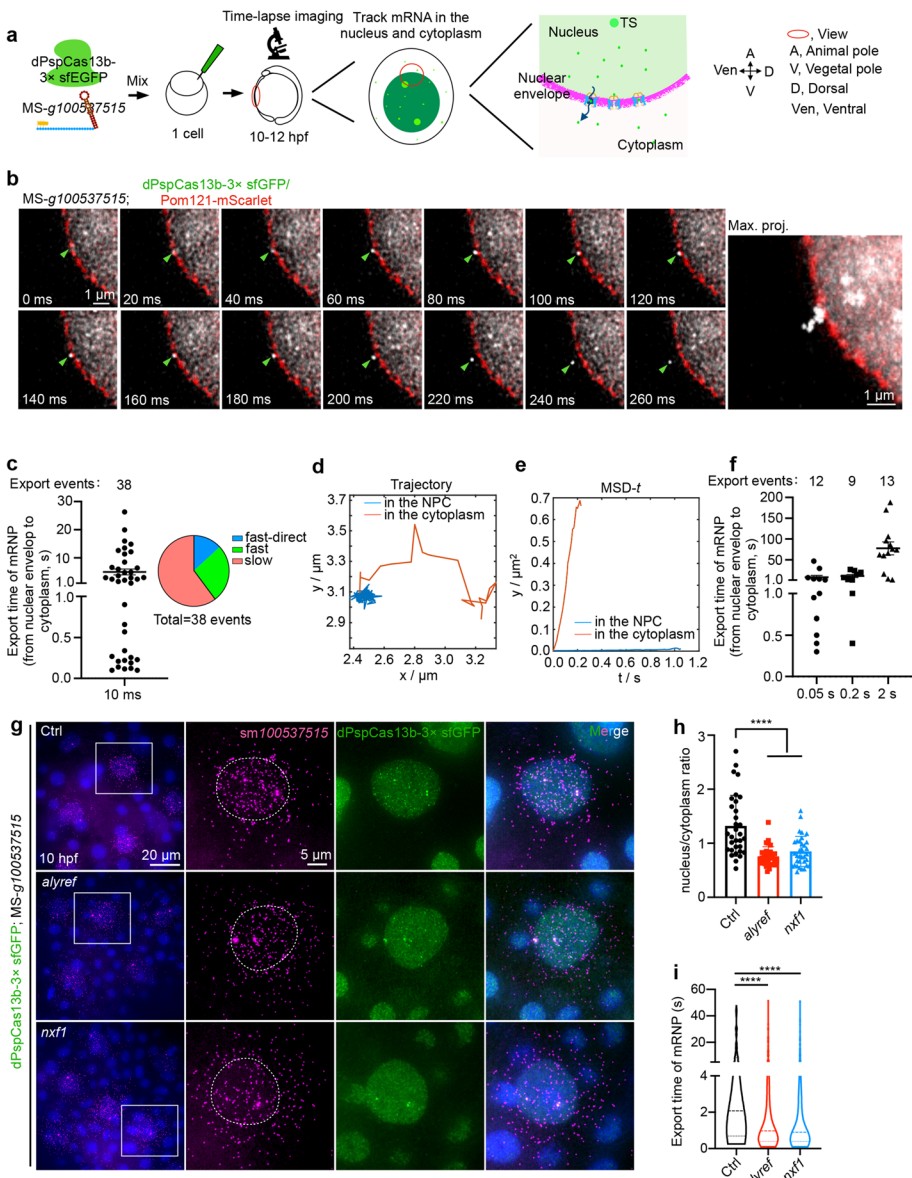

**Fig. 5** (See legend on previous page.)

**Table 1** Different motion of mRNPs in the nucleus and cytoplasm, track time >= 0.2 s (from 17 cells)

| | In the nucleus (total 13,025 tracked mRNPs) | | | | In the cytoplasm (total 2004 tracked mRNPs) | | | |
|---|---|---|---|---|---|---|---|---|
| | Stationary | Corralled | Diffusive | Directed | Stationary | Corralled | Diffusive | Directed |
| D (μm²/s) | 0.0094 | 0.2818 | 0.2839 | 0.1659 | 0.0064 | 0.1405 | 0.1642 | 0.1267 |
| Velocity (μm/s) | / | 0.5921 | 0.6297 | 3.1715 | / | 0.2665 | 0.4051 | 1.4193 |
| Distribution (%) | 2.95 | 49.51 | 47.39 | 0.08 | 29.78 | 44.64 | 25.49 | 0.05 |

/, due to too short movement length and limitation of resolution, measured velocity may be inaccurate; *D*, diffusion coefficients

mRNPs displayed four patterns in the nucleus and cytoplasm, including stationary, corralled, diffusive, and directed diffusion patterns (Table 1; Additional file 1: Fig. S12c, e; Additional file 7: Movies 9-12; see also "Methods").

Each category and its characteristic of mRNP particles is described in Table 1. Of note, a "stationary" pattern indicates a nearly complete immobility of particles in the observed time period, while a "corralled" pattern refers to a confined movement of particles. In brief, most of mRNPs displayed simple diffusion (diffusive) and corralled diffusion (Table 1; Additional file 1: Fig. S12e; Additional file 7: Movies 10,11). Notably, we also captured the direct movement of mRNPs traveling over a long distance (over 2 μm) in the nucleus (Table 1; Additional file 1: Fig. S12e; Additional file 7: Movie 12). Such a motion was previously only detected in the cytoplasm using the MS2-MCP system [6, 37] and molecular beacons [55]. Interestingly, highly confined movements (stationary diffusion) of mRNPs were observed with a much higher frequency in the cytoplasm (29.78%) than in the nucleus (2.95%) (Table 1; Additional file 1: Fig. S12e; Additional file 7: Movie 9). And the diffusion coefficients of stationary, corralled, and diffusive movements were all remarkably decreased in the cytoplasm, compared to those in the nucleus (Table 1; Additional file 1: Fig. S12f). These analyses indicated that some barriers may exist in the cytoplasm to prevent mRNP movement; or, alternatively, that some mRNPs may have arrived at their cytoplasmic destinations, thereby remaining static.

It should be noted that compared to motions of mRNP reporters (such as *lacZ*, *CFP*, and *dystrophin*) via tagging with MS2 or other means in human cells [6, 37, 38, 55], our recorded endogenous mRNPs displayed a higher diffusion coefficient and different distribution patterns in the nucleus and cytoplasm in developing embryos. Further, the technological advance of Multi-Modality Structured Illumination Microscope (Multi-SIM) microscopy has enabled 10 ms per frame resolution, which is 10–100-fold faster than in previous studies [6, 37, 38, 55].

### Dwelling duration of mRNPs at NPCs determines fast and slow export events

Next, to better characterize the export of dPspCas13b/*100537515* mRNPs, we generated mRNP trajectories using maximum time projection of acquired time-lapse movies (Additional file 1: Fig. S12b, down panel). The nuclear localization of dPspCas13b-3× sfGFP was indicative of the inner nuclear boundary, and co-microinjected *pom121-mScarlet* mRNAs were used to label NPCs in developing embryos. The latter allowed us to confirm whether mRNPs docked at and transported through nuclear pores (Fig. 5b and Additional file 1: Fig. S13a, b; Additional file 7: Movies 13,14). We recorded 38 export events and uncovered two types of motion across NPCs: (1) fast transport occurring in less than 1 s, with some particles having a straight traveling trajectory, and (2) slow transport taking longer than 1 s (Fig. 5b, c and Additional file 1: Fig. S13e, f; Additional file 7: Movies 14-16).

The observed duration of fast export events (average 300 ms) was similar to previous studies detected by MS2-MCP-engineered *1/2-mini-dystrophin* mRNPs in human U2OS cells [38], *β-actin* mRNPs in mouse cells [39], and *GFA1* mRNPs in budding yeast [41] (Table 2). We also observed previously undetected direct export events that traveled in a linear trajectory from the nucleus to the cytoplasm via directed diffusion (Additional file 1: Fig. S13c,d, f; Additional file 7: Movie 16). Such direct export events

**Table 2** The time of mRNP export through the nuclear pore to cytoplasm

| Work | Cellular system | mRNA (s) | mRNA size | Labeling system | Nuclear pore transit times | Temporal precision (ms) |
|------|-----------------|----------|-----------|-----------------|----------------------------|-------------------------|
| Ref. [39] | *M. musculus* | *β-actin* | 3.3 kb | MS2-MCP | Average 180 ms | 20 |
| Ref. [38] | *H. sapiens* | *1/2 -mini-Dystro-phin* | 4.8 kb | MS2-MCP | Estimate 0.5 s | 1000 |
| Ref. [56] | *C. tentans* | *hrp36-containing RNAs* | Various | hrp36 | 65 ms - several seconds | 20 |
| Ref. [56] | *C. tentans* | *Balbiani ring 1 and 2 (BR)* | 32–40 kb | hrp36 | Estimate 20 s | 20 |
| Ref. [40] | *M. musculus* | *β-actin* | 3.3 kb | MS2-MCP | Average 12 ms | 2 |
| Ref. [41] | *S.cerevisiae* | GFA1 | 2.2 kb | PP7-PCP | Average 200 ms | 15 |
| This paper | *Danio rerio* | *Fast export (100537515)* | 2.8 kb | CRISPRdPsp-Cas13b | Average 295 ms | 10 |
| This paper | *Danio rerio* | *100537515* | 2.8 kb | CRISPRdPsp-Cas13b | 0.11–26.3 s (average 5 s) | 10 |
| This paper | *Danio rerio* | *100537515* | 2.8 kb | CRISPRdPsp-Cas13b | 0.3 s–46.75 s | 50 |
| This paper | *Danio rerio* | *100537515* | 2.8 kb | CRISPRdPsp-Cas13b | 0.4 s–26.5 s | 200 |
| This paper | *Danio rerio* | *100537515* | 2.8 kb | CRISPRdPsp-Cas13b | 2 s–188 s | 2000 |

were the fastest among all export events recorded in this study with an estimated mean time of translocation being 139 ms $\pm$ 66 ms (see "Methods").

Trajectory analyses revealed the difference between fast and slow export events, depending on the duration of dwelling of particular mRNPs on NPCs, as shown by real-time movies and trajectories (Fig. 5b and Additional file 1: Fig. S13e, f; Additional file 7: Movies 14-16). Among all tracked events, the recorded duration of dPspCas13b/*100537515* mRNP export varied in a wide range from 0.10 to 30 s (Fig. 5c; Table 2). For mRNPs taking longer than 1 s during export, such particles moved with significantly low diffusion coefficients via static diffusion along NPCs (Fig. 5d, e). Some mRNPs appeared even docking on NPCs for dozens of seconds (Additional file 1: Fig. S13b; Additional file 7: Movie 13 as shown 10 s). Further tracking mRNP export events by intervals of 0.05, 0.2, and 2 s for time-lapse imaging revealed a wide range but long resident times occurring during export (Fig. 5f; Table 2; Additional file 7: Movie 17). For example, at the 2 s resolution, the detected export could last 180 s (Fig. 5f; Table 2). It appears that the difference in the imaging time intervals could record different durations of export. This has been shown in the case of *β-actin* mRNPs in mouse cells, in which the export of *β-actin* mRNPs was 10-fold faster at a 2 ms per frame resolution than those reported at a 20 ms per frame [39, 40] (Table 2). In addition, our observations of a 10-ms detection resolution were akin to hrp36-containing mRNPs in *Chironomus tentans* salivary gland cells by microinjecting fluorescently recombinant hrp36 to label endogenous mRNPs with different sizes, which exhibited a wide range duration of export from 65 ms to 20 s at a 20 ms per frame resolution [56] (Table 2). Nevertheless, it appeared that a wide range of times occurred for the export of mRNPs compared with previous studies (Table 2), depending on their size and on imaging resolution (Table 2).

## Nxf1 and Alyref overexpression enhances dPspCas13b/100537515 mRNP export in zebrafish embryos

The unexpected long dwelling time of dPspCas13b/*100537515* mRNPs (Fig. 5c, f) suggests that the formation of large mRNPs reduces their likelihood passing through NPCs. To test this possibility, we performed smFISH to detect *100537515* mRNAs in non-injection controls and dPspCas13b-3× sfGFP/MS-gRNA injected groups. smFISH showed a reduced nucleus/cytoplasm ratio of *100537515* mRNAs in wild-type embryos, compared to embryos bearing dPspCas13b-engaged *100537515* mRNPs at 10 hpf (Additional file 1: Fig. S14a, b). Thus, the large size of dPspCas13b-engaged mRNPs, similar to MS2-MCP tagged mRNPs [38–41], was likely attributed to mRNPs arrested in the nucleus and the wide-range duration of export we observed (Table 2). Such retarded mRNP export could be rescued by ectopic expression of *Alyref* or *Nxf1*, two factors involved in mRNP export, as shown by the reduced nucleus/cytoplasm ratio and reduced *100537515* mRNAs in the nucleus (Fig. 5g, h and Additional file 1: Fig. S14c-e). Notably, the export time of dPspCas13b/*100537515* mRNPs was markedly shortened by ectopic expression of Alyref or Nxf1 (Fig. 5i). However, ectopic expression of other export factors, such as *ddx19a* (*DDX19B*), *gle1* (*GLE1*), *ddx39b* (*DDX39B*), *thoc2* (*THOC2*), *eny2* (*ENY2*), and *pcid2* (*PCID2*), only modestly or barely altered the nuclear/cytoplasmic distribution of dPspCas13b-engaged *100537515* mRNAs (Fig. 5g, h and Additional file 1: Fig. S14c-e), and the expression of *100537515* mRNA was not altered in embryos injected with dPspCas13b-3× sfGFP/MS-gRNA mixture (Additional file 1: Fig. S14f). These results were consistent with the notion that ALYREF is the key component of TREX to license dPspCas13b-engaged mRNPs to be recognized by transport factor NXF1 for mRNA export [57, 58].

## Discussion

### Optimized CRISPR-dCas13 system is a powerful toolkit for direct RNA visualization at the level of single cells in multicellular organisms

Previous studies have shown that the CRISPR-dCas13 system can label RNA in live cells [15–17]. Our study has broadened this application in developing embryos by establishing an easy CRISPR-dCas13 system for tracking both ectopic and endogenous RNAs in a multicellular organism (Figs. 1 and 2). This has enabled us to characterize transcription dynamics (Figs. 3 and 4 and Additional file 1: Fig. S8-11; Additional file 7: Movies 1-8) and mRNP motions (Fig. 5 and Additional file 1: Fig. S12,13; Tables 1 and 2; Additional file 7: Movies 9-17) in a high spatial-temporal resolution, showing this optimized CRISPR-dCas13 system as a powerful toolkit for endogenous RNA visualization in multicellular organisms.

Visualizing and tracking RNAs is critical for understanding their underlying mechanisms in living organisms [1, 59]. Previous studies have used the MS2-MCP system to address these questions in *C. elegans*, *Drosophila* and zebrafish embryos, and *Arabidopsis* [44, 60–64]. However, these studies can be time-consuming, for example, requiring several months to engineer MS2-fused reporters and MCP-EGFP transgenic lines for indirect RNA visualization in zebrafish with the Tol2 transposon system [65]; also, the site-specific genome knock-in using the CRISPR-Cas9 system has still remained a challenge in zebrafish [66]. So far, even in *Drosophila* embryos, only a few studies have allowed the visualization of

transcription dynamics at engineered-endogenous loci [27, 28]. It should also be noted that insertion of dozens of MS2 aptamers into a target RNA can lead to accumulation of RNA decay fragments-containing MS2 [67–69], alerted RNA expression [17], or impacted RNA processing and subcellular localization [70]. It was thus warranted to develop complementary live cell imaging approaches.

Our optimized CRISPR-dCas13 system with chemically modified gRNAs is less time-consuming and eliminates genetic manipulation, making it easier to visualize endogenous mRNAs and track inter-allelic transcription in the same cells in live organisms. By zygotic injection of purified dCas13-fluorescent proteins and modified gRNAs (around 3,400,000,000 dCas13/gRNA molecules per embryo), we achieved single- or dual-color endogenous mRNA visualization at transcription sites (Fig. 2). Moreover, even when the embryos undergo 15 cell cycles of division, around 100,000 dCas13/gRNA molecules were still maintained in each cell, which has a sufficient brightness for visualizing mRNA at 10 hpf (Additional file 1: Fig. S2m). The resulting signals are robust, allowing direct analyses of inter-allelic transcription dynamics from de novo activation to post-mitotic transcriptional re-activation in two continuous cell cycles (Figs. 3 and 4 and Additional file 1: Fig. S8,9); as well as motion and export of mRNPs in developing embryos (Fig. 5a–f and Additional file 1: Fig. S12,13) similar to previous studies for mRNP export in cultured cells [38–40, 56].

### Profiling the single-gene transcription reveals transcriptional memory as a potential extrinsic noise

The MS2-MCP system has been used in profiling single-gene transcriptional profiles [1, 59]; however, dynamics from de novo transcription to post-mitotic transcriptional re-activation in developing organisms have been rarely captured, in particular at endogenous loci. We observed non-synchronized de novo transcription between inter-alleles of both *eppk1* and *100537515* within the same cells in developing zebrafish embryos (Fig. 3 and Additional file 1: Fig. S8). Similar phenomenon of stochastic transcription activation among different cells was also observed after ZGA in zebrafish embryos by using smFISH [71] and molecular beacon [72]. In addition, compared to non-synchronized de novo transcription, we observed rapid post-mitotic re-activation and synchronized re-activation timing of inter-alleles (Fig. 4a–d and Additional file 1: Fig. S9), revealing the inherited transcriptional active state from mother cells as transcriptional memory (Fig. 4e–g). Importantly, the capacity to dissect inter-allelic transcription in individual cells has enabled us to analyze the contribution of intrinsic and extrinsic noise to transcriptional fluctuations [54]. The correlation of inter-allelic transcription output (Fig. 4h and Additional file 1: Fig. S11a) was consistent with previous observations in cultured human cells utilizing MS2-MCP [26]. Increased inter-allelic correlation of transcriptional outputs in both *eppk1* and *100537515* genes during post-mitotic re-activation showed that transcriptional memory prioritized the intrinsic variation after mitosis, which likely resulted in synchronized post-mitotic re-activation (Fig. 4h–n and Additional file 1: Fig. S11).

We propose that transcriptional memory is likely an extrinsic noise and serves as a potential mechanism to modulate non-synchronized de novo transcription to become synchronously re-activated after mitosis in EVL cells in developing zebrafish

embryos (Figs. 3 and 4 and Additional file 1: Fig. S8-11). It should be noted that a subset of transcription factors and chromatin regulators have been proposed to act as mitotic bookmarking factors, which directly bind the chromosome during mitosis and enable the proper activation of genes after mitotic exit, thereby controlling the transcriptional memory propagation throughout mitotic process [52, 53, 73, 74]. It will be of interest to explore additional factors that can regulate the inherence of transcriptional memory in developing zebrafish embryos. During embryogenesis, it is necessary to leverage transcriptional variability caused by stochastic noise to reproducibly establish cell identity and cell fate [75]. Transcriptional memory likely reduces transcriptional noise to facilitate establishment of cell identity in rapidly developing embryos. However, it should be noted that inter-allelic activation examined in the *hnt* and *ush* genes is synchronized in accordance to the gradient of bone morphogenetic protein (BMP) signaling, which is determined by the synchronous cell-to-cell of transcription along the anterior-posterior (AP) axis [27]. Thus, it will be of interest to determine how many genes undergo non-synchronized de novo activation during zebrafish embryogenesis and in other multicellular organisms in future studies.

### Tracking single mRNPs reveals their variable export behaviors in vivo

Although NPCs are presumably crowded for mRNP translocation [76, 77], no significant rate-limiting step for mRNP export via NPCs has been reported previously using single-molecule tracking [39–41] (Table 2). Mor et al. [38] used the MCP-GFP targeted ectopic expression of human dystrophin cDNA and estimated an export duration of 0.5 s, but it was unreasonable to reach this conclusion using 1s per frame resolution for imaging (Table 2). Using fluorescent-labeled Hrp36 to target different sized endogenous mRNPs noted that larger mRNPs likely spend longer time than smaller mRNPs for export [56] (Table 2). With the optimized CRISPR-dCas13 system, we found that the export of dPspCas13b-engaged *100537515* mRNP displayed notably distinct diffusion patterns with variable dwelling time on NPCs (Fig. 5b–f and Additional file 1: Fig. S13c-f; Table 2; Additional file 7: Movies 14-17).

Our observations suggested rate limitations for mRNP export at NPCs. Two possibilities featured in this dPspCas13b system may contribute to this unexpected phenomenon. First, in contrast to native mRNPs, the recruitment of multiple dPspCas13b-3×sfGFP to examined mRNAs has increased the molecular weight of mRNPs. As physical barriers [76], the central channel of NPC is more easily obstructed by large complexes, leading to the observed variable dwelling time at NPCs (Fig. 5b, c, f and Additional file 1: Fig. S13e; Table 2; Additional file 7: Movies 14,15,17). Similarly, the MS2/PP7-engineered-mRNPs are also larger than untagged mRNPs. Other tools to examine the export time of mRNPs are still warranted in the future. Second, NLSs can be bound by karyopherin, which subsequently interacts with NPCs to import cargoes into the nucleus [78]. Thus, NLSs used in the dPspCas13b system may result in decreased export force, thereby retarded mRNP export.

Each NPC consists of several major domains, encompassing the nuclear basket, central channel, and cytoplasmic filaments [79, 80]. Each domain provides a

structural and molecular fundament for the docking, translocation, and releasing of mRNPs during export [31, 77, 81]. With our limited resolution for NPC imaging, we could not discriminate which NPC domain limits the rate of dPspCas13b/*100537515* mRNP export. Altered expression of a number of transport factors with known roles might help to gain insight into this question (Additional file 1: Fig. S14c-f). For example, we likely excluded the cytoplasmic face of NPC was a rate-limiting step for dPspCas13b/*100537515* mRNP export, as overexpression of *Ddx19a* and *Gle1* did not accelerate *100537515* mRNA traveling into cytoplasm in our labeling system (Additional file 1: Fig. S14c-f). However, how exactly Alyref and Nxf1 acted to enhance the nucleocytoplasmic export of dPspCas13b-enagged mRNPs still remains unclear. Future application of the higher-resolution imaging to detect the colocalization among exporting mRNPs, individual transport factors and NPC domains in real time will be of interest to define rate-limiting steps of mRNP export.

## Conclusions

We have established the utility of the zygotic injection of dCas13-fluorescent protein and modified gRNAs to label endogenous mRNA in developing embryos. At this stage, only a small proportion of mRNAs containing at least eight repeated sequences have been visualized. Further, the multiple dPspCas13b-3×sfGFP engaged with the endogenous mRNAs has inevitably brought additional molecular weights that may affect the motion of mRNPs. Future optimization will be needed to visualize RNAs without repeats using multiple efficient gRNAs in cells and developing embryos. One key question is how to design gRNA pools with less off-target side-effect. In this scenario, both the sequence and conformation of the targeted RNA must be taken into consideration, as we have observed that individual gRNAs targeting different regions of *100537515* displayed different labeling efficiencies (Data not shown). Meanwhile, CRISPR-dCas13-gRNA can be further combined with RNA aptamers and foldon-GFP [45, 82] to increase the local brightness and SNR ratio. Nonetheless, the current study will aid the future development of a more robust spatial-temporal system for tracking endogenous RNAs in multicellular organisms.

## Methods

### Zebrafish

Zebrafish (*Danio rerio*) was raised and maintained at 28.5 °C in water system under Zebrafish technology platform of CAS Center for Excellence in Molecular Cell Science. This study was approved by the Ethical Review Committee of CAS Center for Excellence in Molecular Cell Science, Chinese Academy of Sciences (CAS), China. Zebrafish were staged as previously described [50]. The wild type AB strain was used in this study. For each set of experiment, about twenty couples of males and females (AB, 5–18 months old) were randomly selected and crossed to generate embryos. The embryos collected for microinjections came from random parents mating, and 100–1000 embryos were injected for each set of experiment.

### Plasmid construction

To construct the plasmid expressing *GCN4* RNA element for CRISPR-dCas13 system targeting in zebrafish, the zebrafish *β-actin* promoter (cloned from pTol2-zbactin-E2A-mCherry plasmid donated from Weijun Pan Lab, Shanghai Institute of Nutrition and Health, CAS) and the *48× GCN4* sequence were inserted into pcDNA3$^+$ vector, using the T4 DNA ligase (NEB, Cat. No. M0202S). Other plasmids were constructed using Hieff Clone One Step Cloning Kit (Yeasen, Cat. No. 10905ES25) according to the manufacturer's protocol.

To construct plasmids for 6× His-tagged dCas13 proteins expression from *E. coli*, dCas13 ORFs were individually cloned into pET28a$^+$ vector, fused with fluorescent proteins (FP) (EGFP, mScarlet or sfGFP), and a SV40 nuclear localization signal (NLS) in the N-terminus as well as a Nucleoplasmin NLS in the C-terminus was also included in the vector (abbreviated as dCas13-FP). Of note, dPspCas13b-3× sfGFP was fused with SV40 NLS in the N-terminus and NLSs from SV40 and Nucleoplasmin in the C-terminus.

To construct plasmids expressing mRNAs for in vitro transcription (IVT), the coding sequences of *pom121-mScarlet* and *SV40NLS-dPspCas13b/-dRfxCas13d-EGFP-Nucleoplasmin NLS*, as well as cDNA of *ddx19a*, *gle1*, *nxf1*, *alyref*, *ddx39b*, *thoc2*, *eny2* and *pcid2* genes were cloned into the pCS2$^+$ vector containing the SP6 promoter and the SV40 polyadenylation signal, respectively.

To construct plasmid of Fibrillarin (Fbl)-mCherry, the coding sequence was cloned into pEGFP-C1vector.

All oligos used for plasmid constructions are listed in Additional file 2: Table S1.

### dCas13-FP (fluorescent protein) expression and purification

A number of dCas13-FP plasmids, including dPspCas13b-EGFP, dPspCas13b-3× sfGFP, dBba2Cas13b-EGFP, dPba3Cas13b-EGFP, dHgm4Cas13b-EGFP, dHgm6Cas13b-EGFP, dMisCas13b-EGFP, dPguCas13b-EGFP, dRfxCas13d-EGFP, and dRfxCas13d-mScarlet, were individually transformed into the *E. coli* expression strain, the Transetta (DE3) chemically competent cells (Transgen Biotech, Cat. No. CD801), according to the manufacturer's protocol. After transformation, the cells were cultivated at 37℃, 250 rpm for 2 h, followed by transferring into 1 L of LB culture media for further growing at the same condition. Once the absorbance of the culture media reached the OD$_{600}$ around 0.6–0.8, isopropyl β-D-1-thiogalactopyranoside (IPTG) was added to the final concentration of 0.5 mM (GoldBio, Cat. No. I2481C50) to induce protein expression, and cultured at 16℃, 180 rpm for another 18 h.

The next day, cell pellets were collected by centrifugation (5000*g*, 10 min, 4℃), and resuspended in 25 mL lysis buffer (50 mM HEPES, 500 mM NaCl, 2 mM MgCl$_2$, 50 mM Imidazole, 1 mM DTT, 0.5 mM Phenylmethylsulfonyl fluoride (PMSF)). Then, the resuspension was sonicated at 4℃ by high-pressure homogenizer (Ultrahigh pressure cell crusher UH-06; Union-biotech) followed by centrifugation at 10,000 rpm for 45 min at 4℃. After that, the supernatant cell lysates were collected and sterile-filtered through a 0.22-μm polyvinylidene difluoride membrane (Millipore, Cat. No. GSWP04700). The supernatant was then incubated for 10 min with 1 mL Ni-NTA beads in the column

(referred to as 1 column volume, Ni Sepharose 6 Fast Flow, GE healthcare, Cat. No. 17-5318-01) and then flowed through. Next, the Ni-NTA beads were washed twice with 10 column volumes of the lysis buffer and the bound proteins were eluted with 10 column volumes of the elution buffer (50 mM HEPES, 500 mM NaCl, 300 mM Imidazole, 0.01% v/v Triton X-100, 10% glycerol, 1 mM DTT, 0.5 mM PMSF). Proteins were then concentrated using a Amicon® Ultra-15 Centrifugal Filter (50K, Millipore, Cat. No. UFC905008) by centrifugation at 4000$g$ at 4°C and were sterile-filtered before purification by Akta Pure FPLC (GE healthcare). The proteins were further purified through a 5-mL HiLoad Superdex 200 PG of gel filtration chromatography column, which was first equilibrated with storage buffer (50 mM Tris-HCl, 500 mM NaCl, 10% glycerol, 2 mM DTT, pH 7.5). Protein-containing fractions were collected (Additional file 1: Fig. S1c, d) and concentrated, followed by quantification with serial dilutions of standard BSA (2, 1, 0.5, 0.25, and 0.125 µg) by SDS-PAGE gel using Coomassie Brilliant blue staining (Additional file 1: Fig. S1e). Finally, proteins were snap-frozen in liquid nitrogen and stored in aliquots at −80°C.

### RNA synthesis and purification

For the chemically modified gRNAs, including 5′ end 2′-O-methyl 3′ phosphorothioate (MS) modified and/or cyanine 3 (Cy3) labeled, and 3′ end Cy3 labeled gRNAs for dPspCas13b targeting, as well as the 3′ end MS-modified gRNAs for dRfxCas13d targeting, were synthesized at GenScript Company.

For gRNAs produced by IVT, the DNA templates for gRNAs were amplified by primers, followed by agarose gel purification. The gRNA sequences containing the T7 promoter (GAAATTAATACGACTCACTATA) are listed in Additional file 2: Table S1. The IVTs were done with T7 polymerase (Promega, Cat. No. P1300) to produce gRNAs. After that, the gRNAs were purified from denatured PAGE gel.

For mRNAs produced by IVT, DNA templates were digested from constructed pCS2-coding sequence plasmids with NEB restriction endonuclease, either NotI or KpnI, respectively, followed by purification with StarPrep Gel Extraction Kit StarPrep (GenStar, Cat. No. D205-04). mRNAs were then transcribed and purified in vitro by mMESSAGE mMACHINE™ SP6 kit (Thermo Scientific, Cat. No. AM1340) according to the manufacturer's protocol.

### Construction of dCas13-EGFP/gRNA complexes in vitro

To make the in vitro assembled dCas13-EGFP/gRNA complex, we first diluted the IVT or chemically modified gRNAs into 2 µL annealing buffer (10 mM Na-HEPES pH 7.4, 30 mM KCl, and 1.5 mM MgCl$_2$) and annealed the gRNAs by heating at 75°C for 5 min, and slowly cooling down to room temperature at a rate of −0.1 °C/s. Then, each dCas13-EGFP protein was mixed with the annealed gRNAs at the molar ratio of 1:1.5 or 1:3 in the assembly buffer (20 mM Na-HEPES, pH 7.0, 200 mM KCl, 1 mM TCEP) to the final volume of 4 µL, respectively, and incubated at 37°C for 20 min for assembling. The assembled dCas13-EGFP/gRNA complexes were then checked by electrophoretic mobility shift assay (EMSA). Briefly, the 1 µL assembled complexes were loaded onto a native PAGE gel, consisting of 6% acrylamide at the top half and 12% acrylamide at the bottom half, and run the gel in 0.5× TBE buffer at 20 mA for

50 min at 4°C. Then, the proteins of dCas13-EGFP/gRNA complexes were detected by Coomassie Brilliant blue staining and imaging, and the gRNAs were detected by ethidium bromide (EB) staining and imaging (Additional file 1: Fig. S1f, g). The assembled complexes were made to appropriate concentrations for zebrafish zygotic microinjection in different experiments.

To make the dCas13-EGFP/gRNA mixture in vitro without assembly, the modified gRNAs were firstly added into 2 μL annealing buffer, and then mixed with the dCas13-EGFP protein in the assembly buffer to the final volume of 4 μL at room temperature. The final concentrations of the gRNA and dCas13-EGFP in the mixtures for different experiments were shown below in the next section.

The assembled complexes or mixtures were placed on ice prior to microinjection.

### Microinjection of zebrafish embryos

We prepared each corresponding sample detailed below and injected ~1 nL sample into the 1-cell of each embryo at the 1-cell stage. Of note, the concentrations of the CRISPR-dCas13 systems, plasmids, and mRNAs used for microinjection were optimized and had no obvious effect for embryo development.

To screen a panel of dCas13-EGFP proteins for *48× GCN4* RNA labeling (Additional file 1: Fig. S1e), the pre-assembled 5.6 μM dCas13-EGFP/8.4 μM gRNA complexes were used. We either co-microinjected the *β-actin-48× GCN4* plasmid together with the dCas13-EGFP/gRNA complexes or the corresponding dCas13-EGFP protein alone to target *48× GCN4* RNA. For example, 50 pg *β-actin-48× GCN4* plasmid together with 0.9 ng dPspCas13b-EGFP/160 pg gRNA complex or 0.9 ng dPspCas13b-EGFP protein was injected into each embryo, respectively.

For dCas13-EGFP proteins, including dBba2Cas13b, dPba3Cas13b, dHgm4Cas13b, dHgm6Cas13b, and dRfxCas13d, we could hardly detect the EGFP signal in the nucleus post the microinjection of either the complex or the protein alone at the indicated concentration above after 6 h post fertilization (hpf) by widefield microscopy imaging. Tested high concentrations of these pre-assembled dCas13-EGFP/gRNA complexes or the individual dCas13-EGFP proteins alone (i.e., 22.0 μM dPba3Cas13b-EGFP) in the microinjection experiments still yielded poor signals. Of note, for dHgm6Cas13b-EGFP, which was difficult to dissolve and obtain high concentration, 4.5 μM protein/8.4 μM gRNA pre-assembled complex was used in this study.

To label endogenous mRNAs, we microinjected 0.9–1.5 ng dPspCas13b protein/53-160 pg modified gRNA for the CRISPR-dPspCas13b-mediated RNA labeling, and 4–7 ng dRfxCas13d protein/55–160 pg modified gRNA for CRISPR-dRfxCas13d-mediated RNA labeling, either the pre-assembled complex or the mixture. Of note, for the CRISPR-dRfxCas13d system, we found that the low concentration of the protein needed to assemble with high concentration of the modified gRNA to yield the better visualization signal. In our hands, the assembled 0.9 ng dPspCas13b-EGFP/160 pg modified gRNA, or 7 ng dRfxCas13d-EGFP/55 pg modified gRNA were injected into the one-cell stage embryo that yielded reliable SNR after 15 cell cycles in developing embryos.

To label endogenous *muc5.1* and *100537515* mRNAs, respectively, 0.9 ng dPspCas13b-EGFP/160 pg MS-gRNA mixture was injected into each embryo. To achieve

two different endogenous RNAs with dual-color, 0.9 ng dPspCas13b-EGFP/160 pg MS-gRNA and 7 ng dRfxCas13d-mScarlet/160 pg gRNA-MS mixture were co-microinjected into each embryo.

To track endogenous *eppk1* and *100537515* transcriptions, as well as *100537515* mRNP motions, we microinjected 1.5 ng dPspCas13b-FPs/160 pg modified gRNA mixture to each embryo.

To express transport factors, 50 pg *alyref*, 200 pg *ddx19a*, 200 pg *ddx39b*, 200 pg *eny2*, 200 pg *gle1*, 200 pg *pcid2*, 50 pg *nxf1* or 200 pg *thoc2* mRNAs produced by IVT were individually injected into each embryo together with the CRISPR-dPspCas13b system, respectively. Of notes, embryos were developed normally by microinjecting these mRNAs under these tested concentrations. To visualize NPCs, 100 pg *pom121-mScarlet* mRNA together with CRISPR-dPspCas13b system were injected into each embryo. For dCas13-EGFP mRNA injection, 250 pg dPspCas13b-EGFP or 250 pg dRfxCas13d-EGFP mRNA was injected into each embryo. Detailed concentrations of other samples were indicated in the figure legends and methods.

### Widefield microscopy imaging

We checked and collected embryos with relatively uniform fluorescence intensity in examined embryos under stereomicroscope (Nikon SMZ18) before imaging on DeltaVision. Then, live embryos were dechorionated with 1-mL syringe at corresponding developmental stages or fixed embryos after performing smFISH (detailly described below) were mounted on the bottom of the dish (Cellvis, Cat. No. 35-10-1.5-N) with 1% low melting agarose (Thermo Scientific, Cat. No. 16520050). Imaging of embryos was done with DeltaVision Elite imaging system (GE Healthcare) equipped with a 60×/1.42 NA Plan Apo oil-immersion objective. After that, the raw images were deconvolution treated. The deconvolution parameters included enhanced ratio (aggressive) deconvolution, 10 number of cycles, applied correction, normalized intensity, used photosensor, 50% camera intensity offset, and Olympus_60X_142_10612.otf files.

### Screening endogenous RNAs containing repeated sequences

To find endogenous targets of CRISPR-dCas13 system, marker genes that are specifically expressed in each cell of clusters ranging from 4 to 24 hpf were screened from the published single-cell RNA-seq datasets [48]. The pipeline used for identifying the transcripts containing repeated sequences was detailed as below (also referred to Fig. 3a). Firstly, we selected 1383 transcripts, with the longest isoform of each marker gene, as candidates to search short repeated sequences that were used for CRISPR-dCas13 system targeting. Secondly, we searched 20 nt repeated motifs in these transcripts, and compared those fragment sequences to determine mismatches between any two fragments. We then collected the fragment clusters of repeated sequences, in which all fragments matched exactly or had only one mismatch in one of the fragments. Meanwhile, the number and position of the cluster-containing fragments in transcripts were recorded. Thirdly, the cluster-containing fragments presented in more than one gene were removed to achieve unique repeated sequences for CRISPR-dCas13 system targeting. Finally, we selected the fragments having the maximum one mismatch compared to each other within a cluster and used as candidates for labeling.

This computational pipeline generated 134 transcripts containing at least two repeats without overlapping in position Additional file 1: Fig. S4a; Additional file 6: Table S5), among which 15 transcripts contained at least eight repeated sequences (Additional file 1: Fig. S4b). The single-cell count matrices (accession number: GSE112294) were downloaded from Gene Expression Omnibus. The expression of transcripts containing repeated sequences in different cell types as reported in Wagner et al. [48] were calculated as the average of normalized UMI (Unique Molecular Identifiers) counts in each cell (Additional file 1: Fig. S4a, b).

The 134 transcripts with repeated sequences and their expression levels are listed in Additional files 4,5,6: Tables S3,4,5.

### Whole-mount single-molecule RNA fluorescent in situ hybridization (smFISH)

All probes used for smFISH were designed via Stellaris Probe Designer with default parameters (https://www.biosearchtech.com/stellaris-designer) and synthesized in Tsingke biotechnology company.

Probes were labeled with Cyanine 3 (Cy3, *100537515* and *muc5.1* probe) or Red 650 (*GCN4* and *eppk1* probe) at the 3′ ends by Terminal Transferase (NEB, Cat. No. M0315L). In brief, reaction mixture for a 20 μL system includes the following: 1 μL probe mix (stock 100 μM), 2 μL Cy3 or Red 650 (stock 0.2 mM), 2 μL $CoCl_2$, 2 μL enzyme buffer, 0.5 μL Terminal Transferase, and 12.5 μL $ddH_2O$ to make a 20 μL final volume, according to the manufacturer's protocol. Reaction was carried out at 37°C for 4 h and then purified with sodium acetate precipitation.

The procedure of whole-mount smFISH for zebrafish embryos was referred to the previous study with modifications [83]. Fish embryos were fixed in 4% paraformaldehyde (PFA) at 4 °C overnight. The next day, completely removed the fixation solution and washed the embryos twice with 1× PBST (1× PBS and 0.1% Tween-20) for 5 min each, followed by dechorionizing the embryos with 1-mL syringe. The embryos were then dehydrated with 50% methanol/50% 1× PBST once for 5 min and 100% methanol once for 5 min and kept in 100% methanol at −20°C for at least 4 h. After rehydration with 75% methanol/25% 1× PBST, 50% methanol/50% 1× PBST, and 25% methanol/75% 1× PBST one by one (each step took 5 min once), the fixed embryos were then washed twice with 1× PBST for 5 min each and were incubated in 2× SSCT (2× SSC and 0.1% Tween-20) once for 5 min. After that, the embryos were transferred to the prehybridization buffer (10% formamide, 2× SSC, 0.1% Triton X-100) at 30°C and kept for 10 min. Meanwhile, diluted the probe stock solution (5 μM) by the hybridization buffer (10% formamide, 2× SSC, 0.1% Triton X-100, 0.02% BSA, 2 mM Ribonucleoside Vanadyl Complex and 10% dextran sulfate) at the ratio of 1:20. The hybridization was done by incubating the embryos with 100-μL probe at 30°C for overnight (14–16 h). Then, the embryos were washed twice with the prehybridization buffer at 30°C for 30 min each, once with 2× SSCT at 30°C for 30 min, and once with 1× PBST for 5min at room temperature.

For nucleus staining, embryos were incubated with DAPI solution (1:1000, Thermo Scientific, Cat. No. D1306) for 2–5 min and washed twice with 1× PBST for 20 min each. Embryos post smFISH were imaged under the widefield microscopy.

Probe sequences are listed in Additional file 2: Table S1.

### Total RNA isolation, cDNA synthesis and RT-qPCR

Zebrafish embryos at corresponding developmental stages were collected, and the total RNAs from equal number of embryos were extracted with Trizol Reagent (Invitrogen, Cat. No. 15596026) according to the manufacturer's protocol. The cDNA synthesis was carried out using 5× PrimeScript RT Master Mix (TaKaRa, Cat. No. RR036A) according to the manufacturer's protocol. Quantitative (q)PCR was performed using SYBR Green Realtime PCR Master Mix (TOYOBO, Cat. No. QPK-201) and with StepOne-Plus real-time PCR system (Applied Biosystems).

Primer sequences for RT-qPCR are listed in Additional file 2: Table S1.

### Whole mount in situ hybridization (WISH)

The DNA template for the probe targeting *eppk1* contained a T7 promoter sequence (GAAATTAATACGACTCACTATAGGG) and was amplified from cDNA of 24 hpf zebrafish embryos by primers (Additional file 2: Table S1). After purifying with the agarose gel, the *eppk1* probe was transcribed in vitro by T7 polymerase (Thermo Scientific, Cat. No. EP0111) with 10× Digoxigenin RNA Labelling Mix (Roche, Cat. No. 11277073910) according to the manufacturer's protocol and further purified with MEG-Aclear Kit (Thermo Scientific, Cat. No. AM1908). WISH was performed as described previously [84]. In brief, the embryos were fixed in 4% PFA at room temperature for 4 h. After dehydration and rehydration, which were the same as smFISH described above, embryos then were hybridized with *eppk1* probe (1 µg/mL) at 65°C for 16–18 h, followed by incubation with anti-Dig-AP antibody (Roche, Cat. No. 11093274910) at 4°C overnight. The WISH signals were developed in 0.5 mL NBT/BCIP solution (one NBT/BCIP tablet dissolved in 10 mL ddH$_2$O containing 0.1% Tween 20) (Roche, Cat. No. 11697471001), and the embryos were observed and captured with a stereomicroscope (Nikon SMZ18).

### Time-lapse imaging to track transcription and mRNP motion

Before imaging, live embryos at corresponding developmental stages were dechorionated with 1-mL syringe and were mounted on the bottom of the dish with 1% low melting agarose. Embryos were maintained on the equipped live cell imaging chamber at the experimental temperature of 28.5°C.

To track RNA transcription, a serial 3D stack imaging (0.4 µm z-step) in time series was carried out using Olympus SpinSR confocal microscopy with the 60× /1.42 NA UPLXAPO oil-immersion objective and achieved at the 2048 × 2048 pixels field. During tracking *eppk1* transcription in cell cycle 13, due to its relatively short cell cycle period, the image stack was recorded every 2 min. During tracking *eppk1* transcription in cell cycle 14 as well as *100537515* transcription in cell cycles 14 and 15, the image stack was recorded every 5 min. The total tracking time lasted approximately 4 h. Twenty percent 488 nm laser power and 100 ms exposure time were used. Then, a clear image stack in

time series was produced by the maximum intensity projection, and the signals at the transcription sites were tracked and analyzed over time manually.

mRNP motions were tracked with different time series. To track mRNP motion at 10-ms interval, serial 2D image stacks of each cell were acquired at the 512 × 512 pixels widefield by Multi-SIM, developed by Dong Li lab, Institute of Biophysics, CAS). The image stack in time series was collected with 100× /1.49 NA oil objective (Nikon CFI SR HP Apo) and detected by a sCMOS camera (ORCA-Fusion, Hamamatsu) with 80% 488 nm laser power, or with 100% 561 nm laser power (for simultaneous imaging of NPCs and mRNPs), at 2 ms exposure time and 10-ms interval for 30 s (3000 fames). Image stacks in time series were denoised as described in the section "Single-particle tracking" below.

mRNP export events were tracked in different time series. To track mRNP export at 50-ms, 200-ms, and 2-s intervals, different 2D image stacks in time series were acquired by Olympus SpinSR confocal microscopy. One or multiple cells were recorded at different position of the 2048 × 2048 pixels field using a 100× /1.50 NA UPLXAPO oil-immersion objective with 100% 488 nm laser power. Forty nine-millisecond exposure and 50-ms interval for 100 s (2000 fames), 100-ms exposure and 200-ms interval for 100 s (1000 fames), and 100-ms exposure and 2-s interval for 15 min (450 frames) were applied, respectively.

### Imaging processing and analysis

Images of fixed and live embryos were analyzed by Fiji (ImageJ, https://imagej.net/Welcome). Representative images from widefield imaging stacks were performed with maximum intensity projection.

#### *Colocalization analysis*

The signals were selected using straight line and were analyzed by plot profile. We recorded each channel of data and quantified the relative intensity over the distance performed with GraphPad Prism 8.

#### *Quantification of signal-to-noise ratio (SNR)*

SNR was defined as the ratio of the intensity of a fluorescent signal to the power of the background noise. The puncta at the transcription sites were selected with a circle (the diameter of which was 2–3 μm), and the puncta signal was measured with the max intensity. The center of the puncta (exclude the puncta) as background was measured with mean intensity of background. Calculating the SNR with the formula below:

$$\text{SNR} = \left(\text{max intensity of puncta signal} - \text{mean intensity of background EGFP signal}\right)$$
$$/\text{std.dev.of background EGFP signal.}$$

### Detection of mRNP export events

We generated all the mRNPs' trajectories of each cell using maximum time projection from acquired time-lapse movies. The trajectories of export events were detected between the nuclear edge and cytoplasm. We confirmed the exporting events in real time and calculated the time of mRNP export manually. In brief, the nucleocytoplasmic export of mRNPs was firstly observed along the nuclear boundary shown by maximum time projection of dPspCas13b-3× sfGFP or along NPCs labeled by Pom121-mScarlet. Before releasing into cytoplasm, many mRNPs would dwell on NPC region for varying time from milliseconds to minutes. After releasing into cytoplasm, these mRNPs would begin quick diffusion to leave the nuclear edge within two continuous frames, from which we could determine the start and the end of the exporting process. Then we calculated the time of mRNPs moving from the nuclear boundary or NPCs to the cytoplasm. In the cytoplasm, nuclear exported mRNPs could be tracked for about 1 μm or even longer distance, for example shown in Fig. 5b and Additional file 7: Movie 14. To calculate the exporting time of directed export events, we first estimated the length of NPCs, which was about 200 nm (nuclear basket ~75 nm, central framework ~70 nm, and cytoplasmic filaments ~50 nm) [85, 86]. Then, we calculated the velocity of the directed export mRNPs from the single-particle tracking data and estimated the time of directed export events as 139 ms ± 66 ms.

### Quantification of normalized transcription activity

The complicated background, including the movement of EVL cells, other cell type interference, and non-specific aggregation in a fraction of EVL cells (Additional file 7: Movies 1, 2, 5 and 6), made it challenging to extract fluorescent traces of each allele over time with the available algorithm. The maximum intensity projection was performed to produce clear image stack in time series. The signals were identified at the transcriptional locus manually. During cell mitosis, we determined this process by using nuclear morphology as the EGFP fluorescence shown. After re-establishing the clear nucleus, we began to record the signal to measure the re-initial transcriptional activity and observed that most of the *eppk1* and *100537515* allelic re-initial expression occur within 10 min post-mitosis. A puncta region of signals was detected by circle region of interest (ROI-1) to measure the maximum intensity as the puncta transcriptional activity ($ROI\text{-}1_{max}$). A larger circle region (ROI-2, three times the radius than ROI-1) was used surrounding the puncta to measure the mean intensity out of the puncta region as the local background ($ROI\text{-}2_{mean}$). In general, for each data set, the ROI with fixed size was chosen to include the local signal and background throughout all time points. If the puncta signal was too low to be identified, the "puncta signal" was then assigned as the local background signal. Thus, the normalized transcription activity at each punctum and each time point was calculated by the following formula:

Normalized signal intensity = ($ROI\text{-}1_{max}$ − $ROI\text{-}2_{mean}$) / $ROI\text{-}2_{mean}$, which would correct the photobleaching over time. All normalized transcription activities were then plotted against the time/duration.

### Quantification of smFISH signals

To quantify the *100537515* smFISH signals in the nucleus and cytoplasm, we used 3D Object Counter plugin in Fiji (https://imagej.nih.gov/ij/plugins/track/ objects.html).

Briefly, we cropped the image of single EVL cell from raw data with deconvolution. The smFISH signals were identified by the manual bandpass threshold using objects counter 3D, and the number of particles was measured as the total number of *10053751* mRNA signals. Then, the nucleus was chosen based on DAPI signals, and the chosen region was added into ROI manager. After that, we obtained the *10053751* mRNA signals in nucleus and cytoplasm by calculating the smFISH particles.

**The correlation of inter-allelic transcription output and inter-allelic difference expression**

The total output of alleles was the integrated area under the normalized fluorescence trajectory. In details, the normalized signal intensity of puncta1 or puncta2 was cumulated from each observed cell. Then the cumulated fluorescent signals of puncta1 and puncta2 were used for drawing correlation plot in each cell cycle by ggplot (v3.3.4). Pearson correlation coefficient, Spearman's rank correlation coefficient, and slope were calculated by R (v4.1.3, http://www.R-project.org/). For comparison, the cumulated fluorescent signals of puncta1 and puncta2 from randomized cells were paired and used for correlation analysis as described above.

For measurement of the allelic difference expression within a cell, the difference expression was calculated between normalized signal intensity of puncta1 and puncta2 for each observed cell, then the significance levels of difference expression comparison at each time point were calculated from paired two-tailed Student's *t* test. For inter-allelic correlation at each time point, Pearson correlation coefficient was calculated with normalized signals intensity of inter-alleles, and the significance levels of the correlation were calculated from unpaired two-tailed Student's *t* test. To compare inter-allelic correlation of gene's two cell cycles, we collected Pearson correlation coefficient at all time points for each cell cycle; then, *p*-values of the comparison of Pearson correlation coefficient between different cycles were calculated from unpaired two-tailed Student's *t* test. No statistical method was used to pre-determine sample size. No data were excluded from the analyses. All statistical analyses were performed with R package 4.1.1.

**Single-particle tracking**

Live cell images were acquired using the single-molecule tracking mode integrated into the Multi-Modality Structured Illumination Microscope (Multi-SIM) [87]. For each cell, 3000 frames of wide-field images were acquired at the speed of 10 ms per frame. Before single-molecule tracking analysis, the time-lapse images from a total of 17 cells were denoised with the optimization function:

$$\arg \min_{g} \left\| f - H \otimes g \right\|_2^2 + \lambda \left\| g \right\|_1 + \mu \left\| \nabla g \right\|_2^2 \tag{1}$$

where *f* denoted the raw image, *g* denoted the denoised image, *H* denoted the point spread function of the system, $\nabla g$ denoted the 1st-order derivative of denoised image, $\|\cdot\|_n$ denoted the *n*st matrix norm, and $\lambda$ and $\mu$ denoted the weight of each corresponding term.

After denoising, the image stacks were analyzed with the Fiji plugin of TrackMate developed for automated single-particle tracking [88], which generated the position

and time information for all tracks. After filtering out the tracks of less than 20 frames duration, more than 10,000 tracks were collected for motion classification. Next, the tracked data were analyzed with MATLAB (https://www.mathworks.com). First, we computed the mean squared displacement (MSD) for each track, and then fitted with different model functions to classify them into four types of directed, diffusive, corralled, and stationary according to the following protocol.

(1) If the ratio of the smaller to the larger principal radius of gyration was less than 0.001 and the max displacement was longer than 2 μm, then the track was mostly linear, which was classified as directed motion [6, 89]. And its MSD-*t* curve was fitted with:

$$MSD(t) = (vt)^2 \tag{2}$$

where *v* was the linear velocity.

(2) If a trajectory was not directed motion, we fitted its MSD-*t* curve with the following two equations, respectively [90, 91].

$$MSD(t) = 2mD * t^{\alpha} \tag{3}$$

where *t* was lag time, *m* was the dimensionality of image, *D* was diffusion coefficient, *α* was the anomalous exponent.

$$MSD(t) = \frac{L^2}{3}\left(1 - e^{-\frac{t}{\tau}}\right) \tag{4}$$

where *L* was the size of confined microdomains, *τ* represented equilibration time, and its diffusion coefficient *D* was given by $D = L^2/(12\tau)$.

Then we checked which fitting result was better [92]. If Eq. (3) more precisely described the trajectory, the molecule underwent diffusive motion and had few interactions with surrounding components. Otherwise, the molecule was confined into a limited area that was classified as corralled motion. For directed motion, the diffusion coefficients were fitted with Eq. (3), where alpha approached 2.

(3) We found some molecules were classified as diffusive or corralled motion, but their max displacement and diffusion coefficient were quite small. Therefore, we defined such motions as stationary particles if the max displacement and diffusion coefficient of a trajectory were smaller than 400 nm and 0.03 μm²/s, respectively.

### Other quantification analysis

Significant difference was calculated with unpaired two-tailed Student's *t* test, paired two-tailed Student's *t* test, or Mann-Whitney test, and histogram and line chart were

plotted with GraphPad Prism 8. The data was presented as mean ± standard deviation (SD) or mean ± standard error of the mean (SEM) in triplicate experiments, unless otherwise stated; see also figure legends and methods for details. At least two independent experiments were done to gain representative images for microscopy imaging. For the statistical significance and sample size of all graphs, please see figure legends for details.

## Supplementary Information

---

**Additional file 1: Fig. S1.** dCas13 protein purification and assembly with gRNA. **Fig. S2.** Screening the capable dCas13 proteins to label *48× GCN4* RNA. **Fig. S3.** Modified gRNAs enhance the CRISPR-dCas13 labeling on RNAs in zebrafish embryos. **Fig. S4.** Screening the endogenous mRNAs with repeated sequences for labeling. **Fig. S5.** The labeling capacity of dPspCas13b and dRfxCas13d systems on RNA. **Fig. S6.** The simultaneous delivery of dCas13 protein and gRNA enhances the labeling capability. **Fig. S7.** Visualization of other endogenous mRNAs using the CRISPR-dPspCas13b system. **Fig. S8.** Tracking *eppk1* and *100537515* de novo transcription by CRISPR-dPspCas13b. **Fig. S9.** Post-mitotic transcriptional re-activation of *eppk1* and *100537515*. **Fig. S10.** The difference of inter-allele transcriptional activity over time during de novo transcription and post-mitotic re-activation. **Fig. S11.** The correlation of inter-alleles during de novo transcription and post-mitotic transcriptional re-activation. **Fig. S12.** Different motion types of *100537515* mRNPs in the nucleus and the cytoplasm. **Fig. S13.** Different patterns of dPspCas13b/*100537515* mRNP export. **Fig. S14.** Transport factors contribute to dPspCas13b/*100537515* mRNP export.

**Additional file 2: Table S1.** List of oligonucleotides and primer sequences used in this study. (A) Primer sequences used in RT-qPCR. (B) Primer sequences for WISH probe template amplification. (C) Sequences of oligonucleotides used in unmodified gRNA. (D) Sequences of oligonucleotides used in modified gRNA. (E) Sequences of oligonucleotides used in smFISH probes. (F) Primer sequences used in clone.

**Additional file 3: Table S2.** The labeling ability of all examined dCas13 proteins with modified gRNAs.

**Additional file 4: Table S3.** Expression of 134 transcripts containing unique repeated sequences at different development time (Related to Fig. 2a, Additional file 1: Fig. S4a).

**Additional file 5: Table S4.** Expression of 15 transcripts containing at least eight repeated sequences at different development time (Related to Fig. 4b).

**Additional file 6: Table S5.** The information of 134 transcripts including repeated sequence, number of repeated sequences, position of repeated sequence and length of transcripts.

**Additional file 7.** Movies. The recorded dynamic transcription and mRNP motion in this study.

**Additional file 8.** Review history.

---

### Acknowledgements

We would like to thank G. Carmichael for reading the manuscript and discussion. We thank Guang Xu, Lin Shan, Peng-Fei Luan, Xiao-Qi Liu, and Ying Wang for help in experiments and in analyses, and other in Chen and Yang laboratories members for the discussion.

### Review history

The review history is available as Additional file 8.

### Peer review information

### Authors' contributions

L.-L.C. conceived the project. Y.-K.H. and L.-L.C. designed experiments. Y.-K.H. performed experiments with the help of L.-Z.Y., Y.-H.P., H.W.; B.-Q.G. and X.-K.M. performed computational analyses, supervised by L.Y.; Q.M. performed single-particle tracking, supervised by D.L.; Y.-K.H., H.W., and L.-L.C. drafted the manuscript and wrote the manuscript with the input from all authors. L.-L.C. supervised the project. The author(s) read and approved the final manuscript.

### Funding

This work was supported by the National Key R&D Program of China (2021YFA1100203), the CAS Project for Young Scientists in Basic Research (YSBR-009), the Shanghai Municipal Commission for Science and Technology (20JC1410300), the National Natural Science Foundation of China (NSFC) (31830108, 31725009, and 31821004), the Center for Excellence in Molecular Cell Science (CEMCS) (2020DF03), and the HHMI International Program (55008728) to L.-L.C.; NSFC 31925011 and 2019YFA0802804 to L.Y.; NSFC 32125024 and 2021YFA1300303 to D.L.; and China Postdoctoral Science Foundation (2021M693165) and Shanghai Municipal Fund for Daily Expenses (E15C6032) to Y.-K.H. L.-L.C. acknowledges the support from the XPLORER PRIZE.

### Availability of data and materials

All data generated or analyzed during this study are included in this published article and its supplementary information files. The single-cell RNA-seq datasets of zebrafish embryos were downloaded from NCBI GEO (GSE112294) [48].

## Declarations

**Ethics approval and consent to participate**
Not applicable.

**Consent for publication**
Not applicable.

**Competing interests**
The authors declare that they have no competing interests.

**Author details**
[1]State Key Laboratory of Molecular Biology, Shanghai Key Laboratory of Molecular Andrology, CAS Center for Excellence in Molecular Cell Science, Shanghai Institute of Biochemistry and Cell Biology, University of Chinese Academy of Sciences, Chinese Academy of Sciences, 320 Yueyang Road, Shanghai, China. [2]CAS Key Laboratory of Computational Biology, Shanghai Institute of Nutrition and Health, University of Chinese Academy of Sciences, Chinese Academy of Sciences, Shanghai, China. [3]National Laboratory of Biomacromolecules, CAS Center for Excellence in Biomacromolecules, Institute of Biophysics, Chinese Academy of Sciences, Beijing, China. [4]College of Life Sciences, University of Chinese Academy of Sciences, Beijing, China. [5]Center for Molecular Medicine, Children's Hospital, Fudan University and Shanghai Key Laboratory of Medical Epigenetics, International Laboratory of Medical Epigenetics and Metabolism, Ministry of Science and Technology, Institutes of Biomedical Sciences, Fudan University, Shanghai, China. [6]School of Life Science and Technology, ShanghaiTech University, Shanghai, China. [7]School of Life Science, Hangzhou Institute for Advanced Study, University of Chinese Academy of Sciences, Hangzhou, China.

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

## 