## [**Additional file 8.** Review history. · Genome Biology]

Review History

First round of review

Reviewer 1

Are you able to assess all statistics in the manuscript, including the appropriateness of statistical tests used? Yes, and I have assessed the statistics in my report.

Comments to author:

The authors introduced an optimized toolkit for tracking developmentally expressed mRNAs in zebrafish embryos based on catalytically dead CRISPR-Cas13 (dCas13)-fluorescent proteins and modified guide (g)RNAs, allowing direct analysis of de novo transcription and post-mitotic re-activation between inter-alleles and mRNP movement in zebrafish embryos. This paper is excellent and provides an expanded application of the CRISPR-dCas13 system in developing embryos for direct RNA visualization. Minor points to consider in subsequent versions:

1. Line 146: The size of one of the puncta shown by the white arrow in Fig. 1b is not significantly different from the size of the aggregation in Extended Data Fig. 2f,g. How can signal puncta be distinguished from EGFP aggregates in living cells? The signal puncta in lower left of Fig. 1b (white arrows shown) seems large. In fixed cells, smFISH can be used to validate the signal puncta, however, how to correctly identify the signal puncta in living cells?
2. Fig. 3I: The size of the red dots pointed by the arrow is much larger than the green one, how to distinguish whether these signals are "puncta" or "aggregation"?
3. Some sentences contain grammatical mistakes, such as an extra ", " before "(3,4)" in Line 54.
4. Fig 4c shows only one example, it will be helpful to add more statistics on when de novo transcription starts: average, range, etc.
5. If Fig4 was obtained by confocal microscopy, it is helpful to list more details of imaging acquisition and processing, e.g. how many stacks acquired and the step distances.
6. Fig5. Have the authors compared the time of post-mitotic transcriptional re-activation between eppk1 and 100537515? It is interesting to know if post-mitotic transcriptional re-activation of different genes happen at a similar rate or at different rates.
7. Fig 5f-g legend. typo for "re-activation".

Reviewer 2

Are you able to assess all statistics in the manuscript, including the appropriateness of statistical tests used? Yes, and I have assessed the statistics in my report.

Comments to author:

The authors have established the CRISPR-dCas13 system for tracking ectopic and endogenous RNAs in developing embryos of Zebrafish. By zygotic infection of purified dCas13-fluorescent proteins and modified gRNAs, they achieved single- or dual-color imaging of endogenous mRNAs in Zebrafish embryos which undergo 15 cell cycles division from ZGA. This RNA monitoring system was then applied for direct analyses of transcription dynamics and mRNP export in developing embryos. They found non-synchronized de novo transcription between inter-alleles and synchronized post-mitotic re-activation in pairs of alleles. Moreover, dCas13-engaged mRNPs moved rapidly within the nucleus, but

varied widely in export times throughout NPCs. Ectopic expression of Alyref or Nxf1 could promote nucleocytoplasmic export of such mRNPs.

There are currently no appropriate methods for imaging endogenous RNAs in a multicellular organism. This work presents the first report to utilize the CRISPR-dCas13 systems for tracking endogenous mRNAs in developing organisms. As such, I believe this paper will be an important contribution to the field. However, the following points should be addressed prior to publication.

Major comments:

1. Chemically modified gRNAs significantly improved the RNA labeling efficiency of CRISPR-dPspCas13b and CRISPR-dRfxCas13d. What about other dCas13 systems? Are there more options of the dCas13 systems that use chemically modified gRNAs for RNA labeling?
2. While this Reviewer appreciates the analysis in Fig. 1 (b-d), I think that these results probably should go in Supplementary information. Because the labeling efficiency is very low and dCas13-GFP was not co-localized well with smFISH.
3. Although chemically modified gRNAs improve dCas13-mediated RNA labeling, there are very few dCas13-GFP spots representing the proportion of mature RNAs in the cytoplasm. In addition to the SNR in Fig. 2f, the number of dCas13-GFP spots in the nucleus and cytoplasm needs to be quantified separately. The colocalization analysis of dCas13-GFP and smFISH would provide a better indication of the labeling efficiency of the dCas13-GFP system.
4. In Fig. 5, the inheritance of active transcription states has been analyzed during cell division. However, it is not clear in Fig. 5a when cells undergo mitosis. H2B-mCherry could be included to improve the quality of images and analyses.
5. Lines 400-402, "These findings indicated that the transcriptional memory reduced transcriptional fluctuations after mitosis, leading to more synchronous transcription between post-mitotic inter-alleles." Synchronous post-mitotic transcription is likely contributed by the transcriptional memory. However, this is only a correlation based on the observation in this study.
6. It is intriguing that dCas13b-3xsfGFP/100537515 mRNP export is modulated by Alyref and Nxf1. Is the export time of mRNA significantly affected by the ectopic expression of Alyref and Nxf1? In Fig. 6g, mRFPs distribution in the nucleus and cytoplasm were indicated by smFISH. What about dCas13-GFP labeling here?
7. The authors could not exclude the effects of NLSs and dCas13-3xsfGFP tagging on the dynamics of mRNP export. This should be discussed thoroughly.

Minor comments:

1. This manuscript could be written a little more cohesively.
2. Lines 54-56, "methods for understanding RNA processing at a high spatial and temporal resolution in living cells, in particular, in vivo are still limited." This work has mainly focused on the studies of transcription dynamics and mRNA export, so it is not appropriate to mention RNA processing here.
3. In the legend of Fig. 3j, 'ns' represents 'not significant' or 'no significant difference'.

Reviewer 3

Are you able to assess all statistics in the manuscript, including the appropriateness of statistical tests used? Yes, and I have assessed the statistics in my report.

Comments to author:

This study exploits the catalytic dead Cas13 RNA imaging system for RNA tracking in zebrafish embryos. The authors thoroughly examined the stability of multiple Cas13 family proteins, the binding affinities of sgRNAs with chemical modifications, and the stoichiometry between dCas13 protein and sgRNAs. Through optimization, the CRISPR-dCas13 system can be used to visualize repetitive sequences containing endogenous gene transcription in vivo.

Consistent with previous studies, the study identifies the non-synchronized behavior between alleles. In addition, the authors also examined the RNA export dynamics. However, the findings do not represent a significant advance over previous published works.

1. In Figure 2b and movies 1, 5, and 6, additional transcription sites like signal spots are observed in the nuclei, bringing the concern of off-targeting labeling of the dCas13-sgRNA system. Therefore, it would be more instructive if the authors could identify the population's off-target rates.
2. The diffraction limited spot of the transcription-site should be fitted with the Gaussian function.
3. The dCas13 complex was introduced with microinjection. It is not clear how many different embryos were analyzed here and what is the signal spectrum across individual embryos.
4. When calculating postmitotic transcription activities, which timepoint served as time zero? As two daughter cells may have different nuclear envelope re-assemble kinetic, is this a factor correlated with differences in transcription activity?
5. Is there a consistent number of how many fluorescent molecules are bound per RNA transcript? Without this prior knowledge, it is premature to use the fluorescent intensity to refer to the single molecules.
6. dCas13 contains NLS. The author should consider the effect caused by NLS in the mRNP export assay. Or more specifically, does the system suitable for mRNP export analysis?

Point-by-point response to each reviewer:

We thank all the reviewers for their constructive and helpful comments on our manuscript that have guided us to include additional data, analyses and discussions to further improve our work. We have addressed all points raised by the reviewers, please read our point-to-point responses below in blue. Of note, the summary of all changed panels was listed in the table below; and all changes in the revised manuscript were marked in blue.

Summary of all changed panels including the revised figures, Extended Figures and Supplementary Table.

Major changes to figure contents and Table		
Figures in the original submission (Old figures)	Figures in this submission (New figures)	Changes
Fig. 1b, c and Extended data Fig. 2f, g	Extended data Fig. 2a, f, g, i (revised)	White arrows replaced with magenta arrows
Fig. 5d	Fig. 4d (revised)	Unpaired two-tail Student's t test was included
Fig. 6g	Fig. 5g (revised)	The colocalization of dPspCas13b-3× sfGFP and sm100537515
New	Fig. 5i	The export time of dPspCas13b-3× sfGFP/ 100537515 mRNPs by ectopically expressed Ayref or Nxf1
New	Extended data Fig. 2j, k	- The 2D-size of signal puncta and EGFP aggregates -The colocalization of EGFP aggregates with Fbl-mCherry
New	Extended data Fig. 3g, h	dPguCas13b labeling 48× GCN4 RNA in live and fixed embryos
New	Extended data Fig. 5d-f	Spots of transcription sites were fitted with Gaussian function
New	Extended data Fig. 5q, r	- The labeling efficiency of CRISPR-dPspCas13b system for eppk1 - The off-target ratio of CRISPR-dPspCas13b system for eppk1
New	Extended data Fig. 7m, n	- The labeling efficiency of CRISPR-dPspCas13b system for 100537515 - The off-target ratio of CRISPR-dPspCas13b system for 100537515
New	Extended data Fig. 8h	Time of de novo expression of eppk1 and 100537515

Major changes to figure contents and Table		
New	Extended data Fig. 9e	Time of transcriptional re-activation between eppk1 and 100537515
Extended data Fig. 9e	Extended data Fig. 9f (revised)	Unpaired two-tail Student's t test was included
New	Extended data Fig. 12a	The number of dPspCas13b-EGFP molecules bind to single 100537515 mRNA
New	Table S2	The labeling capability of all examined dCas13 proteins with modified gRNAs
Major changes to figure order		
Figures in the original submission (Old figures)		Figures in this submission (New figures)
Fig. 1b-e		Extended data Fig. 2a, i, l, m
Fig. 2		Fig. 1b-i
Fig. 3		Fig. 2
Fig. 4		Fig. 3
Fig. 5		Fig. 4
Fig. 6		Fig. 5
Extended data Fig. 2a-g		Extended data Fig. 2b-h
Extended data Fig. 5d-m, n, o		Extended data Fig. 5g-p, s, t
Extended data Fig. 9e-g		Extended data Fig. 9f-h
Extended data Fig. 7a, b		Extended data Fig. 7e, f
Extended data Fig. 12a-e		Extended data Fig. 12b-f

Reviewers' Comments:

Reviewer #1: The authors introduced an optimized toolkit for tracking developmentally expressed mRNAs in zebrafish embryos based on catalytically dead CRISPR-Cas13 (dCas13)-fluorescent proteins and modified guide (g)RNAs, allowing direct analysis of de novo transcription and post-mitotic re-activation between inter-alleles and mRNA movement in zebrafish embryos. This paper is excellent and provides an expanded application of the CRISPR-dCas13 system in developing embryos for direct RNA visualization. Minor points to consider in subsequent versions:

Response: We thank the reviewer for the positive comments and constructive suggestions on our work. We have addressed all the remaining points as detailed below.

Comments

1. Line 146: The size of one of the puncta shown by the white arrow in Fig. 1b is not significantly different from the size of the aggregation in Extended Data Fig. 2f,g. How can signal puncta be distinguished from EGFP aggregates in living cells? The signal puncta in lower left of Fig. 1b (white arrows shown) seems large. In fixed cells, smFISH can be used to validate the signal puncta, however, how to correctly identify the signal puncta in living cells?

Response: We thank the reviewer for raising these questions. We apologize for raising this unnecessary confusion by using the same color of arrow heads to show EGFP aggregates and signal puncta in the previous Fig. 1b (now Extended Data Fig. 2a). We have changed white arrows to magenta arrows to indicate EGFP aggregates in the revised Extended Data Fig. 2a,f,g (previous Fig. 1b and Extended Data Fig. 2f,g). Of note, the magenta arrow in lower left of Extended Data Fig. 2a (previous Fig. 1b of white arrow) indicates EGFP aggregates but not signal puncta. Indeed, our new data have shown that these EGFP aggregates are localized to nucleoli (new Extended Data Fig. 2j,k), which have been described in the revised manuscript on page 5 (lines 145-156).

2. Fig. 3I: The size of the red dots pointed by the arrow is much larger than the green one, how to distinguish whether these signals are "puncta" or "aggregation"?

Response: We thank the reviewer for raising this concern. We assumed that the question is about the results of Fig. 3I (not Fig. 3i). In the previous Fig. 3I (now Fig. 2I), the red dots pointed by the arrowheads were two adjacent transcription spots of signal puncta labeled by dRfxCas13d-mScarlet. These red dots are larger than the green ones (the middle panel) because only one dPspCas13b-EGFP-labeled spot at the *muc5.1* locus was observed. As the non-specific EGFP aggregates were localized in nucleoli (new Extended Data Fig. 2k), which are nine-fold larger than signal puncta detected by dCas13s (new Extended Data Fig. 2j), signal puncta can be well separated from EGFP aggregations throughout our experiments. We have now clearly defined this point on page 5 of the revised manuscript.

3. Some sentences contain grammatical mistakes, such as an extra ", " before "(3,4)" in Line 54.

Response: Thank you, we have carefully checked the manuscript and fixed mistakes and grammar problems.

4. Fig 4c shows only on example, it will be helpful to add more statistics on when de novo transcription starts: average, range, etc.

Response: We thank the reviewer for this suggestion. Our analyses indeed have included all examined cells (n= 55 cells for *eppk1*; n= 24 cells for *100537515*) of dynamic transcription in real-time during *de novo* transcription (Extended Data Fig 8 f,g). As suggested, we have included a new panel that indicates when *de novo* transcription starts for *eppk1* and *100537515* (new Extended Data Fig. 8h). Accordingly, we have revised the manuscript on page 8 (lines 281-282, 293-294).

5. If Fig4 was obtained by confocal microscopy, it is helpful to list more details of imaging acquisition and processing, e.g. how many stacks acquired and the step distances.

Response: We thank the reviewer for this suggestion. The previous Fig. 4 (now Fig. 3) was obtained by confocal microscopy, and we have included imaging acquisition and processing in details in the Fig. 3 legend on page 28 (lines 984-988) of the revised manuscript.

6. Fig5. Have the authors compared the time of post-mitotic transcriptional re-activation between *eppk1* and *100537515*? It is interesting to know if post-mitotic transcriptional re-activation of different genes happen at a similar rate or at different rates.

Response: We thank the reviewer for raising this question. We set the time of zero minute as the initiation of transcriptional re-activation as soon as the re-establishment of a clear nucleus, indicated by the nuclear GFP fluorescence background signals. Comparison of the initial time of post-mitotic transcriptional re-activation between *eppk1* and *100537515* showed that the post-mitotic transcriptional re-activation of both genes began at a similar rate in different cells (new Extended Data Fig. 9e). Consistently, a similar rate of transcriptional re-activation was also detected in the same cells of intra-alleles between *eppk1* and *100537515* loci (revised Fig. 4d and Extended Data Fig. 9f). These analyses indicated that transcriptional re-activation of different genes may occur at a similar rate. We have included this description on page 9 (lines 334-337).

7. Fig 5f-g legend. typo for "re-activation".

Response: Thank you and fixed.

Reviewer #2: ===

The authors have established the CRISPR-dCas13 system for tracking ectopic and endogenous RNAs in developing embryos of Zebrafish. By zygotic infection of purified dCas13-fluorescent proteins and modified gRNAs, they achieved single- or dual-color imaging of endogenous mRNAs in Zebrafish embryos which undergo 15 cell cycles division from ZGA. This RNA monitoring system was then applied for direct analyses of transcription dynamics and mRNP export in developing embryos. They found non-synchronized *de novo* transcription between inter-alleles and synchronized post-mitotic

re-activation in pairs of alleles. Moreover, dCas13-engaged mRNPs moved rapidly within the nucleus, but varied widely in export times throughout NPCs. Ectopic expression of Alyref or Nxf1 could promote nucleocytoplasmic export of such mRNPs. There are currently no appropriate methods for imaging endogenous RNAs in a multicellular organism. This work presents the first report to utilize the CRISPR-dCas13 systems for tracking endogenous mRNAs in developing organisms. As such, I believe this paper will be an important contribution to the field. However, the following points should be addressed prior to publication.

Response: We thank the reviewer for positive comments and constructive suggestions on our work. We have addressed all the points in detail as described below.

Major comments:

1. Chemically modified gRNAs significantly improved the RNA labeling efficiency of CRISPR-dPspCas13b and CRISPR-dRfxCas13d. What about other dCas13 systems? Are there more options of the dCas13 systems that use chemically modified gRNAs for RNA labeling?

Response: We thank the reviewer for raising this suggestion. Accordingly, we have tested the labeling ability of other dCas13 systems for labeling $48\times GCN4$ RNA using chemically modified gRNA targeting *GCN4* (*gGCN4*). We found although dRfxCas13d and dPguCas13b failed to label $48\times GCN4$ RNAs with an unmodified gRNA (Extended Data Fig. 2f,h), the gRNA at the 3' end spacer modification has enabled RNA detection by dRfxCas13d and dPguCas13b both in live and fixed embryos (Extended Data Fig. 3e,f and new Extended Data Fig. 3g,h; Table S2). Of note, though, dPguCas13b failed to label endogenous *eppk1* mRNAs (Table S2). We have included a new Additional file_Table S2 and revised the manuscript on pages 6 (lines 185-188) and 7 (lines 214-215) accordingly.

2. While this Reviewer appreciates the analysis in Fig. 1 (b-d), I think that these results probably should go in Supplementary information. Because the labeling efficiency is very low and dCas13-GFP was not co-localized well with smFISH.

Response: We thank the reviewer for this suggestion. The previous Fig. 1 (b-e) have been moved as the Extended Data Fig. 2. And the previous Fig. 1a and Fig. 2 have been combined as the new Fig. 1 in the revised submission.

3. Although chemically modified gRNAs improve dCas13-mediated RNA labeling, there are very few dCas13-GFP spots representing the proportion of mature RNAs in the cytoplasm. In addition to the SNR in Fig. 2f, the number of dCas13-GFP spots in the nucleus and cytoplasm needs to be quantified separately.

Response: We thank the reviewer for raising this suggestion. In the previous Fig. 2f (now Fig. 1g), the ectopic $48\times GCN4$ RNAs were only visualized in the nucleus, we could not further quantify the number of dCas13-GFP spots in the cytoplasm. As these

48×GCN4 RNAs were ectopically expressed, we think that it is not necessary to quantify the number of *48×GCN4* in the nucleus. Instead, we have quantified the tracked spots of endogenous *100537515* mRNPs (Extended Data Fig. 12f, and Table 1), which have been labeled by the dCas13 systems in both nucleoplasm and cytoplasm in live embryos.

The colocalization analysis of dCas13-GFP and smFISH would provide a better indication of the labeling efficiency of the dCas13-GFP system.

Response: To characterize the labeling efficiency of the dCas13-EGFP system, we analyzed the colocalization patterns of the endogenous *eppk1* and *100537515* mRNAs between signals detected by dCas13-EGFP and smFISH approaches. For detected transcription sites of these loci, dPspCas13b worked efficiently (> 90%) and had no off-target (new Extended Data Fig. 5q,r and 7m,n). For those detected by the single-molecule *100537515* mRNA FISH, the labeling efficiency is about 47.3% with an about 4.1% off-target rate (new Extended Data Fig. 7m,n). We have included these results on pages 7 (lines 233-235) and 8 (lines 267-270).

4. In Fig. 5, the inheritance of active transcription states has been analyzed during cell division. However, it is not clear in Fig. 5a when cells undergo mitosis. H2B-mCherry could be included to improve the quality of images and analyses.

Response: We thank the reviewer for raising this suggestion. It is true that H2B-mCherry is a standard experiment to track daughter cells during cell division. However, although the nuclear EGFP intensity was decreased during mitosis, the remaining EGFP intensity in the cytoplasm was still sufficiently visible, and has enabled us to track two daughter cells divided from a mother cell, according to the nuclear morphology illustrated by EGFP fluorescence (Rebuttal_Fig. 1, see an example with green arrowheads). Thus, we could indeed visualize this process by examining the nuclear morphology shown with the EGFP fluorescence. We have included this description in the revised manuscript on page 9 (lines 324-326).

Rebuttal_Fig. 1: dPspCas13b-EGFP is localized into the nucleus in living fish embryos. During cell division, the low EGFP intensity was still visible and could indicate cell division as green arrowheads shown. Scale bar: 10 μ m. Time shows hour: minute (h: min)

5. Lines 400-402, "These findings indicated that the transcriptional memory reduced transcriptional fluctuations after mitosis, leading to more synchronous transcription between post-mitotic inter-alleles." Synchronous post-mitotic transcription is likely

contributed by the transcriptional memory. However, this is only a correlation based on the observation in this study.

Response: We thank the reviewer for raising this question and we agree. Additional evidence is needed to support the transcriptional memory-mediated synchronous post-mitotic transcription. Since zygotic gene transcription events in early embryogenesis can be influenced by many factors, including pioneer transcription factors, RNAP II machinery and chromatin regulators (*Dev Cell* 2014, PMID: 24780732; *Development* 2017, PMID: 29042475), redistribution of some of these factors, and other unknown changes may make the nuclear environment more favorable for transcriptional re-activation between inter-alleles after cell division. While studying these mechanisms in detail goes beyond the scope of the current study, we have softened the description of this result as to “*transcriptional memory may serve as a potential mechanism in mediating synchronous post-mitotic transcription*” in the revised manuscript on pages 10 and 11.

6. It is intriguing that dCas13b-3xsfGFP/100537515 mRNP export is modulated by Alyref and Nxf1. Is the export time of mRNA significantly affected by the ectopic expression of Alyref and Nxf1? In Fig. 6g, mRFPs distribution in the nucleus and cytoplasm were indicated by smFISH. What about dCas13-GFP labeling here?

Response: We thank the reviewer for raising these questions. To assess the effect of Alyref and Nxf1 on the export time of dCas13b-3xsfGFP/100537515 mRNP, we have recorded export events of dPspCas13b-3xsfGFP/100537515 mRNPs at 50 ms per frame resolution upon the ectopic expression of Alyref and Nxf1. The new results showed that the export time of mRNPs was shortened by expression of Alyref or Nxf1 (new Fig. 5i). In the meanwhile, we have included the colocalization of dCas13-GFP and sm100537515 in the revised Fig. 5g (previous Fig. 6g), and revised the manuscript accordingly on page 14 (lines 514-515).

7. The authors could not exclude the effects of NLSs and dCas13-3xsfGFP tagging on the dynamics of mRNP export. This should be discussed thoroughly.

Response: We thank the reviewer for raising this suggestion and we agree. During the revision, we have attempted to examine the effects of NLSs fused to dCas13-3xsfGFP tagging on the dynamics of mRNP export by generating the dPspCas13b-3xsfGFP systems with different copy and strength of NLSs (Rebuttal_Fig. 2). Our results showed that the CRISP-dPspCas13b system could not efficiently label 100537515 mRNAs with fewer NLSs or NLSs with less strength in the nucleus (Rebuttal_Fig. 2), compared to the NLS-dPspCas13b-3xsfGFP-NLS-NLS (v1) used (Rebuttal_Fig. 2b, left panel). Nonetheless, we've included a discussion on the effect of NLSs on mRNPs export in the revised manuscript on page 17 (lines 625-628).

Rebuttal Fig. 2: The effect of nuclear localization signals (NLSs) on CRISPR-dPspCas13b system labeling *100537515* mRNAs. (a) dPspCas13b-3xsfGFP variants with different number, position and strength of NLSs were constructed. Of note, the v1 version containing two SV40 NLS and one nucleoplasmin NLS have been used in the current study. The nucleoplasmin NLS is stronger than the SV40 NLS for nuclear localization at our hands (data not shown). Other three types of dPspCas13b-3xsfGFP (v2, v3 and v4) contain fewer copy number or less strong NLSs. (b) The reduced strength of NLSs led to failed labeling of *100537515* mRNAs by the dPspCas13b-3xsfGFP system in transcription sites and in the nucleoplasm.

Minor comments:

1. This manuscript could be written a little more cohesively.

Response: We thank the reviewer for raising this suggestion. We have shortened the manuscript to make it more cohesively in the revised submission, including changes on pages 4-5, 7-13, 16-17, 44.

2. Lines 54-56, "methods for understanding RNA processing at a high spatial and temporal resolution in living cells, in particular, in vivo are still limited." This work has mainly focused on the studies of transcription dynamics and mRNA export, so it is not appropriate to mention RNA processing here.

Response: We thank the reviewer for raising this suggestion. We have corrected this description in the revised submission in lines 60-62.

3. In the legend of Fig. 3j, 'ns' represents 'not significant' or 'no significant difference'.

Response: Thank you and we have corrected this label in the legend of Fig. 2j (previous Fig. 3j).

====

Reviewer #3: ===

This study exploits the catalytic dead Cas13 RNA imaging system for RNA tracking in zebrafish embryos. The authors thoroughly examined the stability of multiple Cas13 family proteins, the binding affinities of sgRNAs with chemical modifications, and the stoichiometry between dCas13 protein and sgRNAs. Through optimization, the CRISPR-dCas13 system can be used to visualize repetitive sequences containing endogenous gene transcription in vivo.

Consistent with previous studies, the study identifies the non-synchronized behavior between alleles. In addition, the authors also examined the RNA export dynamics. However, the findings do not represent a significant advance over previous published works.

Response: We thank the reviewer for the comments on this study. Although similar live-cell imaging approaches, especially the MS2-MCP system, have been used to track dynamic RNA export and non-synchronized behavior between alleles in cultured cells, our work broadens the application of CRISPR-dCas13 for endogenous RNA labeling and contributes to the field for understanding the dynamics and motions of untagged gene transcription and mRNP export in developing vertebrate embryos.

1. In Figure 2b and movies 1, 5, and 6, additional transcription sites like signal spots are observed in the nuclei, bringing the concern of off-targeting labeling of the dCas13-sgRNA system. Therefore, it would be more instructive if the authors could identify the population's off-target rates.

Response: We thank the reviewer for raising this concern. Although EGFP aggregates were observed in the previous Fig. 2b (now Fig. 1c) and movies 1, 5 and 6, we could separate non-specific aggregates well from real signals, as such non-specific aggregates were enriched to the nucleoli (new Extended Data Fig. 2k) and were much larger (ie. 9-fold larger than signal puncta) (new Extended Data Fig. 2j). We have described these differences in the revised manuscript on page 5 (lines 150-152).

Because EGFP aggregates could be distinguished from real signals in live embryos and were nearly undetectable in fixed embryos with smFISH (for example, Fig. 2e), we thus could examine the off-target rate by comparing these images. We found that for mRNAs detected by the dPspCas13b system at transcription sites, the labelling efficiency is > 90% without detectable off-target (new Extended Data Fig. 5q,r and 7m,n). However, detection of the single-molecule mRNAs has remained a challenge. Compared to smFISH results for the same mRNAs, the labeling efficiency of single-molecule *100537515* mRNA is ~ 47.3% with 4.1% off-target rate in the nucleoplasm (new Extended Data Fig. 7m,n), while a much lower labeling of cytoplasmic *100537515* mRNAs (Data not shown). We have now included the labeling efficiency and off-target rate of the CRISPR-dPspCas13b system for both *eppk1* and *100537515* mRNAs labeling on pages 7 (lines 233-235) and 8 (lines 267-270).

2. The diffraction limited spot of the transcription-site should be fitted with the Gaussian function.

Response: We thank the reviewer for raising this question. We have examined the diffraction limited spot of the transcription-site fitted with the Gaussian function, which showed ~ 0.9 (R^2) of Goodness of Fit (new Extended Data Fig. 5d-f). We've included this data in revised manuscript on pages 6-7 (lines 215-217).

3. The dCas13 complex was introduced with microinjection. It is not clear how many different embryos were analyzed here and what is the signal spectrum across individual embryos.

Response: We thank the reviewer for raising these questions. Because the dCas13 complex was introduced by manual microinjection, we would check and collect embryos with relatively uniform fluorescence intensity in examined embryos under stereomicroscope (Nikon SMZ18) before imaging. Detailed description has been included in Methods on page 40 (lines 1276-1278) of the revised manuscript.

In all micrographic images in live and fixed embryos, we usually recorded and analyzed 10-20 embryos for each experiment. The signal spectrum across individual embryos has been illustrated in Extended Data Fig. 2m, in which comparable EGFP intensity across individual embryos at different development stages was shown. In addition, we have also included the detail for the number of embryos recorded and analyzed in each panel legend.

4. When calculating postmitotic transcription activities, which timepoint served as time zero?

Response: We thank the reviewer for raising this question. We determined the process of cell division using the EGFP fluorescence-indicated nuclear morphology (see Rebuttal_Fig. 1 as an example). As soon as the re-establishment of the clear daughter nuclei, we set the time zero for the initiation of transcriptional re-activation and began to record the intensity of signal to measure the transcriptional activity. We have included this detail in revised manuscript on page 9 (lines 324-326).

As two daughter cells may have different nuclear envelope re-assemble kinetic, is this a factor correlated with differences in transcription activity?

Response: We thank the reviewer for raising this question, but we feel that addressing this question is out of the scope of our study. The different nuclear envelope re-assemble kinetic maybe correlated with differences in transcription activity when the asymmetric nuclear division happens to daughter cells (*Curr Biol* 2021, PMID: 34297912). However, we did not observe significantly different size of nuclei between daughter cells according to our time-lapse imaging during cell division (Rebuttal_Fig. 1); in the meanwhile, we observed that the transcriptional re-activation between

daughter cells becomes more synchronized after cell division (Fig. 4 and Extended Data Fig. 9-11).

5. Is there a consistent number of how many fluorescent molecules are bound per RNA transcript? Without this prior knowledge, it is premature to use the fluorescent intensity to refer to the single molecules.

Response: We thank the reviewer for raising this concern. By fitting with Gaussian function, the result revealed ~11 copies of dPspCas13b-EGFP per mRNA (new Extended Data Fig. 12a), which is similar to the predicted 12 target sites for dPspCas13b-EGFP binding, indicating that most mRNA is indeed single-molecule labelling. We have included this new result to describe that single *100537515* mRNA bound the number of dPspCas13b-EGFP molecules on page 12 (lines 419-422).

6. dCas13 contains NLS. The author should consider the effect caused by NLS in the mRNA export assay. Or more specifically, does the system suitable for mRNA export analysis?

Response: We thank the reviewer for raising this question and we agree. We have tried to study the effect of NLSs in dPspCas13b-3×sfGFP on mRNAs export, but the results showed other versions of dPspCas13b-3×sfGFP could not efficiently label *100537515* mRNAs (Rebuttal_Fig. 2), compared to the 3x NLSs (v1) used in the current study (Rebuttal_Fig. 2b, left panel). Although this multiple NLSs-containing CRISPR-dCas13 showed the retarded mRNAs export, which might not truly reflect the motion and transport of native mRNAs, this model could still be used to explore which transport factors would affect nuclear export (Fig. 5g-i and Extended Data Fig. 14c-f). We have included a discussion on the possible effect of NLSs on mRNA motions in the revised manuscript on page 17 (lines 625-628).

===

Second round of review

Reviewer 2

The additional work by the authors strengthened the manuscript. I have no further request.

Reviewer 3

The study by Huang et al. took advantage of the CRISPR-dCas13 for endogenous RNA (containing repetitive sequence) imaging. This study improved the SNR in the CRISPR-dCas13 system and expanded the endogenous RNA imaging toolbox.

The author addressed most of the concerns in the revised version.

Some additional minor points:

1. Many works use MS2 imaging in multicellular organisms. Such as PMID: 35508662, PMID: 30038397, PMID: 36243734 etc.
2. Zebrafish has genome duplication. How many transcription sites (theoretically) exist for the genes imaged in this work?
3. It should be cautious about using the current NLS-dCas13 system to study RNA export.
4. For Fig3c, Fig4b, and extended Fig9b, it would be great to show the entire time trace instead of eliminating 4hrs.
5. For Extended Fig7 J and Fig8 C, if the author explained the red-only signal as in-inefficiently targeted RNA, what about the GFP-only spots?

Authors Response

Point-by-point responses to the reviewers' comments:

Reviewer #2: The additional work by the authors strengthened the manuscript. I have no further request.

Response: We thank this reviewer for the positive comments on our work.

Reviewer #3: The study by Huang et al. took advantage of the CRISPR-dCas13 for endogenous RNA (containing repetitive sequence) imaging. This study improved the SNR in the CRISPR-dCas13 system and expanded the endogenous RNA imaging toolbox.

The author addressed most of the concerns in the revised version.

Response: We thank this reviewer for the positive comments on our work. Below please read our point-to-point responses.

Comments

Some additional minor points:

1. Many works use MS2 imaging in multicellular organisms. Such as PMID: 35508662, PMID: 30038397, PMID: 36243734 etc.

Response: We thank the reviewer for the careful reading of our manuscript. We have now included papers (PMID: 35508662, PMID: 30038397) that use MS2 imaging in multicellular organisms in the revised manuscript on page 3. However, the paper (PMID: 36243734) is not related to the using MS2 imaging in multicellular organisms, we thus did not include it in the revised manuscript.

2. Zebrafish has genome duplication. How many transcription sites (theoretically) exist for the genes imaged in this work?

Response: We thank the reviewer for raising this question. When tracking the dynamic transcription, we recorded two distinguishable signal spots (transcription sites) within one allele in close proximity during the late stage of the measured cell cycle (shown by green arrowheads in Additional file 1: Fig. S8d). Thus, in theory, four transcription sites should be imaged for each gene that have 2 alleles, as shown in Fig. 2e. We have briefly mention this point the in the Fig. 2e legend.

3. It should be cautious about using the current NLS-dCas13 system to study RNA export.

Response: We thank the reviewer for raising this concern. We fully agree. Due to the potential recruitment of large protein complexes to the exported RNAs and the effect of NLSs that may change the motion pattern of recorded exporting events, it should be cautious to use the NLS-dCas13 system to study RNA export. We have discussed this concern in details in Discussion on page 17.

4. For Fig3c, Fig4b, and extended Fig9b, it would be great to show the entire time trace instead of eliminating 4hrs.

Response: We thank the reviewer for raising this suggestion. We have updated the Fig3c, Fig4b, extended Fig.S9b and Fig.S8e with the entire time trace to show the dynamic transcription.

5. For Extended Fig7 J and Fig8 C, if the author explained the red-only signal as in-inefficiently targeted RNA, what about the GFP-only spots?

Response: We thank the reviewer for raising this concern. The red-only signal was CRISPR-dCas13b system in-inefficiently targeting *100537515* mRNA, whereas the GFP-only spots might be caused by unknown aggregation of dPspCas13b-GFP proteins, or by single molecule probes in-inefficiently targeting these dPspCas13b/*100537515* mRNPs. We have mentioned these in the Additional file 1: Fig. S7J legend when such GFP-only spots first appeared.